

# A 1/16° eddying simulation of the global NEMOv3.4 sea ice-ocean system

Doroteaciro Iovino[1], Simona Masina[1,2], Andrea Storto[1], Andrea Cipollone[1], and Vladimir N. Stepanov[1]

[1]*Centro Euro-Mediterraneo sui Cambiamenti Climatici (CMCC), Bologna, Italy*
[2]*Istituto Nazionale di Geofisica e Vulcanologia (INGV), Bologna, Italy*

## Abstract

Analysis of a global eddy-resolving simulation using the NEMO (version 3.4) general circulation model is presented. The model has 1/16° horizontal spacing at the equator, employs two displaced poles in the Northern Hemisphere, and uses 98 vertical levels. The simulation was spun up from rest and integrated for 11 model years, using ERA-Interim reanalysis as surface forcing. Primary intent of this hindcast is to test how the model represents upper ocean characteristics and sea ice properties.

Numerical results show that, overall, the general circulation is well reproduced, with realistic values for overturning mass and heat transports. Analysis of the zonal averaged temperature and salinity, and the mixed layer depth indicate that the model average state is in good agreement with observed fields. Comparisons against observational estimates of mass transports through key straits indicate that most aspects of the model circulation are realistic. As expected, the simulation exhibits turbulent behaviour. The spatial distribution of the sea surface height variability from the model is close to the observed pattern. Despite the increase in resolution, the variability amplitude is still weak, in particular in the Southern Ocean. The distribution and volume of the sea ice are, to a large extent, comparable to observed values.

Compared with a corresponding coarse-resolution configuration, the performance of the model is significantly improved, although relatively minor weaknesses still exist. We conclude that the model output is suitable for broader analysis to better understand upper ocean dynamics and ocean variability at global scales. This simulation represents a major step forward in the CMCC global ocean modelling, and constitutes the groundwork for future applications to short-range ocean forecasting.

## 1. INTRODUCTION

The global ocean is a highly turbulent system over a wide range of space and time scales. Both satellite and in situ data show that mesoscale eddies pervade the ocean at all latitude bands. Eddies usually account for the peak in the kinetic energy spectrum and most of their energy is generated and maintained by baroclinic instabilities of large-scale flows. Those processes play a substantial role in the dynamics of the global ocean, e.g., transporting and mixing temperature and salinity, exchanging energy and momentum with the mean flow, controlling the mechanisms of deep water spreading and deep convection preconditioning, and modulating air-sea interactions (see e.g. Morrow and Le Traon 2012). The dominant length scale of these eddies varies greatly with latitude, stratification and ocean depth. Mesoscale eddies typically have horizontal scales of the order of the first baroclinic Rossby radius of deformation, varying



roughly from 200 km in the tropics to 10-20 km at 50-60° (Chelton et al. 1998), vertical scales
ranging from the pycnocline depth to the full ocean depth, and time scales of weeks and
months.
Global numerical ocean models, with spatial resolutions ranging from hundred down to a few
kilometres, often include both regions where the dominant eddy scales are well resolved and
regions where the model resolution is too coarse for eddies to form and hence eddy effects
have to be parameterized. In the context of ocean modelling, a model will be eddy rich as long
as it uses a horizontal grid mesh whose resolution is fine enough to explicitly (albeit partially)
resolve baroclinic and barotropic instability processes, i.e. the grid spacing is finer than the first
baroclinic Rossby radius of deformation. Since the milestone paper by Smith et al. (2000),
eddy effects are considered explicitly modelled when the horizontal grids are refined to at least
1/10° (ca. 12 km); however such resolution compares to the Rossby radius and adequately
describes both mesoscale variability and western boundary currents only for latitudes lower
than 50°. Resolving mesoscale eddy variability remains elusive at higher latitudes (Hallberg
2013). For example, in the Arctic Ocean where the first Rossby radius decreases down to few
kilometres, typical eddy-resolving resolution does only permit eddies at best (Nurser and
Bacon 2014).
A key weakness of nearly all global ocean models used to study climate is the absence of an
explicit representation of ocean mesoscale eddies, since their spatial scale is smaller than the
scale typically resolved by model horizontal grid meshes. Furthermore, operational
oceanography for a variety of different applications such as search-and-rescue, fisheries, and
oil spill requires global ocean forecasting systems to reach kilometric scales in coastal areas.
This demand is also fostered by the continuous increase of resolution in numerical weather
prediction models (Le Traon et al. 2015) and the design of next-generation satellite altimetry
missions that will aim to better capture the ocean mesoscale variability.
These considerations motivate the push toward fully mesoscale eddying ocean models, where
the full dynamics and life cycle of baroclinic eddies can be realistically represented over the
entire global domain. Thanks to progress in ocean modelling and the advances in high
performance computing resources over the last decade, oceanic mesoscale eddying numerical
simulations are now a realistic choice to bring new insights into the physical processes
operating in the ocean and to find application in Earth system modelling and forecasting.
During the last decade, an extensive effort has been made to simulate eddying ocean, different
models have been implemented in regional, near-global and fully global domains (e.g. Maltrud
and McClean 2005, Chassignet et al. 2009, Oke et al. 2013, Drakkar Group 2014, Metzger et
al. 2014, Dupont et al. 2015). In this context, we developed a global eddying configuration,
where eddying means that the numerical simulation is eddy-resolving over all (most of) the
domain. This manuscript seeks to present the general characteristics of an 11-year spin-up
simulation, hereunder called GLOB16, at 1/16° (ca. 6.9 km) equatorial resolution, which is
performed using the state-of-the-art modelling framework NEMO (Nucleus for European
Modelling of the Ocean). The numerical model is a coupled ocean/sea ice model, including a
three-dimensional, primitive equation ocean general circulation model and a dynamic-



thermodynamic sea ice model. So far, GLOB16 represents the NEMO global configuration
having the highest horizontal resolution, and is the first step in the development of a new,
operational short- term ocean forecast system meant to serve as the backbone for downscaling
coastal and regional applications to develop services for the global coastal ocean.
The paper is organized as follows. Section 2 describes the model setup, while model analysis is
found in Section 3. We rely on comparisons with observations, as well as with a twin eddy-
permitting experiment, called GLOB4, as a means of assessing the quality of GLOB16
solution. Conclusions follow in Section 4.
## 94  2. MODEL CONFIGURATION
GLOB16 is a global, eddying configuration of the ocean and sea ice system based on version
3.4 of the NEMO ocean model (Madec et al. 2012). The ocean component OPA is a finite
difference, hydrostatic, primitive equation ocean general circulation model, with a free sea
surface. The ocean component is coupled to the Louvain-la-Neuve sea Ice Model (LIM2)
(Fichefet and Maqueda 1997). The ice dynamics are calculated according to external forcing
from wind stress, ocean stress and sea surface tilt and internal ice stresses using C grid
formulation (Bouillon et al. 2009). The elastic-viscous-plastic (EVP) formulation by Hunke
and Dukowicz (1997) is used. The key features of the configuration follow in this section,
while a comprehensive technical description of GLOB16 is given in Iovino et al. (2014).
### 106  2.1 Mesh
GLOB16 makes use of a non-uniform tripolar grid, computed at CMCC following the semi-
analytical method of Madec and Imbard (1996). The horizontal grid has 1/16° resolution at the
equator, corresponding to 6.9 km, that increases poleward as cosine of latitude, leading to 5762
x 3963 grid points horizontally. The grid consists of an isotropic Mercator grid from 60°S to
20°N. The meridional scale factor is maintained constant at 3 km south of 60°S. The location
of the geographical South Pole is conserved and the domain extents southwards to 78°S,
including the ice shelf edge in the Weddell and Ross Seas. North of 20°N, the grid consists of a
non-geographic quasi-isotropic grid. To avoid singularities associated with the convergence of
meridians at the North Pole, two distinct poles are introduced, whose locations are such that the
minimum horizontal resolution is ~2 km around Victoria Island. Ocean and sea ice are on the
same horizontal grid. The vertical coordinate system is based on fixed depth levels and consists
of 98 vertical levels with a grid spacing increasing from approximately 1 m near the surface to
160 m in the deep ocean.
### 122  2.2 Bathymetry
The GLOB16 bathymetry is generated from three distinct topographic products: ETOPO2
(U.S. Department of Commerce 2006) is used for the deep ocean, GEBCO (IOC, IHO and
BODC 2003) for the continental shelves shallower than 300 m, and Bedmap2 (Fretwell et al.



2013) for the Antarctic region, from 60ºS. The result is modified by two passes of a uniform
Shapiro filter, and finally hand editing is performed in key areas. The Black Sea is connected
to the Marmara Sea through a 1-grid-point wide channel. The Caspian Sea is all derived from
ETOPO2. The maximum depth allowed in the model is 6000 m, the minimum depth is set to
10 m.  Bottom topography is represented as partial steps (Barnier et al. 2006).

### 2.3 Parameterisations
In our simulation, a linearized free surface formulation is used (Roullet and Madec 2000) and a
free-slip lateral friction condition is applied at the lateral boundaries. Biharmonic viscosity and
diffusivity schemes are used in the horizontal directions in the equations of momentums and
tracers, respectively. The values decrease poleward as the cube of the grid cell size. Tracer
advection uses a total variance dissipation (TVD) scheme (Zalesak 1979). Vertical mixing is
achieved using the TKE turbulent closure scheme (Blanke and Delecluse 1993). Background
coefficients of vertical diffusion and viscosity represent the vertical mixing induced by
unresolved processes in the model. Vertical eddy mixing of both momentum and tracers is
enhanced in case of static instability. The turbulent closure model does not apply any specific
modification in ice-covered regions. A diffusion bottom boundary layer parameterization is
used for tracers.

### 2.4 Initialisation
The simulations is started from a state of rest in January 2003, with initial conditions for
temperature and salinity derived from the 1995-2004 decade of the World Ocean Atlas 2013
set of climatologies (WOA13; Locarnini et al. 2013, Zweng et al. 2013). The initial conditions
for the sea ice (ice concentration, ice thickness) correspond to mean January 2003 produced by
a global ocean reanalysis run at 1/4º horizontal resolution (Storto et al. 2015).

### 2.5 Forcing
Forcing fields are provided from ERA-Interim global atmospheric reanalysis (Dee et al. 2011),
released by European Centre for Medium Range Weather Forecasts (ECMWF), with 0.75°
spatial resolution. The turbulent variables are 3 hourly and radiative and freshwater fluxes are
daily. The surface boundary conditions are prescribed to the model using the bulk formulae
proposed by Large and Yeager (2004). The forcing routine and the ice model are called every 4
time-steps (ca. every 13 minutes). A monthly climatology of coastal runoff is derived from Dai
and Trenberth (2002) and Dai et al. (2009), with a global annual discharge of ~1.32 Sv (1 Sv =
$10^6$ $m^3$ $s^{-1}$), and is applied along the land mask. The fresh water is added to the surface,
assumed to be fresh and at local sea surface temperature. As the thickness of the uppermost
level is 0.4 m, diurnal cycle is imposed on solar flux: the daily averaged short wave flux is
spread over the day according to time and geographical position (Bernie et al. 2007). The mean



sea level is free to drift. Shortwave penetration is applied through the RGB (Red Green Blue)
formulation that splits the visible light into three wavebands. The penetration is modulated by a
constant chlorophyll value.
**2.6 Restoring and spin-up**
To avoid drifts in salinity and eventual impacts on the overturning circulation, the sea surface
salinity (SSS) is restored toward the monthly objective analyses from the EN4 data set of the
Met Office Hadley Centre (Good et al. 2013), with a time scale of 300 days for the upper 50 m.
The sea surface temperature (SST) is restored towards the NOAA Optimum Interpolation 1/4°
Daily Sea Surface Temperature Analysis (Reynolds et al. 2007) with a constant damping term
of 200 W $m^{-2}$ $K^{-1}$, which corresponds to a restoring time of 12 days. The restoring is identical
for the open sea and ice-covered areas.
The time step was set to 20 sec for the first 3 days of the simulation, and then increased
progressively to reach 200 sec at the $60^{th}$ day. The model run for 11 years through the end of
2013, which appears to be a sufficient amount of time for the near-surface velocity field to
adjust to the initial density field and for mesoscale processes in the upper ocean to have
reached a quasi-equilibrium, while the deep ocean takes much longer to reach steady state.
This simulation may be therefore appropriate for studying the dynamics of the ocean
circulation on short time scales, but may not for studying the long-term evolution of deep-
water masses or climate variability. GLOB16 experiment was performed using 4080 CPU
cores on an IBM System x iDataPlex supercomputer. Per simulated year, it required 112000
CPU hours and generated ~3 Tb of output files.
**2.7 Ouput and analysis strategy**
For comparison purposes, we performed a twin experiment, GLOB4, at eddy-permitting
resolution (1/4° at the equator), which is detailed in the Appendix. It employs same numerical
schemes and parameterizations as GLOB16, except for the resolution-dependent parameters,
such as the horizontal viscosity and diffusivity, sea-ice viscosity, and the time-step length.
Model outputs are archived as successive 5-day means throughout the whole integration and
post-processed to monthly and annual means. The first simulated year, 2003, is disregarded
because of the initial model adjustment; variability in time is analysed over the period 2004-
2013, while mean values are computed over the last five years of integrations, from 2009 to
2013, unless otherwise indicated.
**2.8 Eddy-permitting configuration**
The eddy-permitting GLOB4 is based on version 3.4 of NEMO (Madec et al. 2012). The
configuration is a global implementation on an ORCA-like tri-polar grid (Barnier et al. 2006).
The horizontal grid, known as ORCA025, has 0.25° resolution (1442 grid points × 1021 grid
points) at global scale decreasing poleward. The effective resolution is ~27.75 km at the
equator, and increases as the cosine of latitude with minima of 3.1 km (5.6 km) in the
meridional (zonal) direction. The model has 75 vertical levels where the level spacing
increases from 1 m near the surface to 200 m at 6000 m. The bathymetry used in GLOB4 is
based on the combination of GEBCO in coastal regions and ETOPO2 in open-ocean areas. A
uniform Shapiro filter is applied twice, and hand editing is performed in a few key areas.
Bottom topography is represented as partial steps.
The model uses a linear free surface and does conserve total energy for general flow and
potential enstrophy for horizontally non-divergent flow. The horizontal viscosity is bi-
Laplacian with a value of -1.8 $\times$ 10$^{11}$ m$^4$ s$^{-1}$ at the equator, reducing polewards as the cube of
the maximum grid cell dimension. Tracers are advected using a total variance dissipation
(TVD) formulation. Lateral diffusivity for tracers is parameterized by a Laplacian operator
with an eddy diffusivity coefficient of 300 m$^2$ s$^{-1}$ at the equator, decreasing polewards
proportionally to the grid size. Vertical diffusion is parameterized by the turbulent kinetic
energy (TKE) scheme. Unresolved vertical mixing processes are represented by a background
vertical eddy diffusivity of $1.2\times10^{-5}$ m$^2$ s$^{-1}$, and a globally constant background viscosity of 1.2
$\times$ 10$^{-4}$ m$^2$ s$^{-1}$. Bottom friction is quadratic. A diffusive bottom boundary layer scheme is
included. GLOB4 has sea ice component, atmospheric forcing, bulk formulation and tracer
restoring in common with GLOB16.

**3. MODEL VALIDATION**
The main objective of this section is to present an overview of the characteristics of the
GLOB16 simulation, evaluate its quality against recent observations and highlight the effect of
eddying resolution against the eddy-permitting run.
The spin-up of the circulation, as measured by the total kinetic energy (TKE, defined as 0.5(u$^2$
+ v$^2$) where u and v are the 5-day averages of the horizontal velocity components), potential
temperature and salinity averaged over the whole domain, is shown in Fig. 1, and demonstrates
the extent to which a quasi-steady state has been reached at the end of the simulation. The TKE
of the system increases rapidly during the first simulated year (2003, not shown) and
approaches ~12 cm$^2$ s$^{-2}$ at the beginning of 2004, indicating a baroclinic adjustment of the
velocity field to the initial density field. Then, the kinetic energy fluctuates between 11.5 and
12.5 cm$^2$ s$^{-2}$ for the rest of the simulation, with the highest contribution given by the Southern
Ocean (Fig. 1a). Most of the kinetic energy is in the eddy field: the mean GLOB16 eddy
kinetic energy (EKE, computed from the 5-day velocity fields using the equation 0.5(u'$^2$ + v'$^2$),
where primes denote deviations from the annual-mean velocities, (u', v') = (u, v) − (<u>, <v>))
contributes by ~56% to the total basin-averaged budget (Fig. 1b). As a result of the increased
resolution, the time mean of the TKE does not change much over the whole basin (being ~10%
larger than in the twin GLOB4 run), while the eddy contribution is boosted by 40% by the
eddying resolution.
As expected, in the spin-up stage of the integration, the model adjusts from the WOA13 initial





conditions towards the new state imposed by the forcing fields and parameter choices. Both
basin-mean potential temperature and salinity show a drift with a clear annual cycle (Fig. 1c,d):
temperature decreases by ~0.01 ºC, while salinity presents a small increase of 0.0013 psu over
the 10-year period.

**3.1 Mean temperature and salinity**

The mean fields of modelled potential temperature and salinity are here validated against
observations. Figure 2 (a, b) show the SST and SSS biases, relative to the EN3 (the UK Met
Office Hadley Centre observational dataset, Ingleby and Huddleston 2007) climatology, both
averaged over the same period 2009-2013. The global mean biases are negative and small (-
0.06 for SST and -0.04 for SSS). There are weak cold biases in the tropics, extending over
much of the subtropical band. The largest SST biases are warm (over 1 °C), and are collocated
with the positive SSS error (0.5–1.5 psu) over the western boundary currents in the Atlantic
and North Pacific oceans. In the Arctic, probably related to biases of air temperature and
radiations in the atmospheric forcing (Barnier et al. 2006), there is a positive (negative) surface
salinity error of up to 2 psu, where there is an excessive sea ice formation (melting).
The surface biases of models forced by prescribed surface boundary conditions are, to a large
degree, constrained by the forcing fields, but the analysis of subsurface fields allow for a
stronger test of the model, revealing discrepancies in diapycnal mixing and advection
pathways. The time- and zonal-average of modelled potential temperature and salinity are
shown in Fig. 2 (c, d), along with their differences from EN3 (Fig. 2e,f). GLOB16 temperature
field reproduces the expected large-scale features (Fig. 2c), with cold waters over all depths at
high latitudes and warm water at shallow, low latitudes. GLOB16 salinity also follows
expectation (Fig. 2d): the low salinity tongue (34.6 psu) of Antarctic Intermediate Water
(AIW), which sinks to ~1500 m depth between 60º-50ºS and propagates toward the equator; an
high salinity (up to 35.2 psu) cell centred around 25ºS over the upper 300 m layer; a surface
salinity minimum of 34.2 psu at 5º-10ºN connected to the strong precipitation in the inter-
tropical convergence zone; high-salinity tongue associated with the Mediterranean Sea at about
35ºN; low-salinity water over the top 200 m north of 45ºN related to the Arctic melt water; and
high-salinity (35.2 psu) water below 300 m depth north of 60ºN associated with the formation
of cold, dense waters in the North Atlantic. All of these features are clearly present in the
observation-based climatology (not shown).
The difference field for temperature (Fig. 2e) indicates that the modelled ocean is generally too
warm at intermediate depth (100-300 m), with the exception of the AIW, colder by 0.4 ºC. The
largest differences, propagating down to 1000 m, are located in the northern hemisphere from
~40ºN (likely due to the Mediterranean Sea) poleward. The locations of the convective site set
the positive and negative biases within the band 60-75ºN. Compared to EN3 temperature, the
upper Arctic Ocean in GLOB16 is too warm (up to ~1.4 ºC at ~300 m), mainly due to a
warmer Barents Sea inflow. The salinity field reproduced by GLOB16 differs from
observations by ~0.15 psu at the most (Fig. 2f). Modelled and observed salinities agree well off



Antarctica. The model is saltier by 0.1 psu at about 50º S in the upper 400 m of the water
column, and by 0.15 psu at the Equator at ~150 m. The model is too saline (up to 0.1 psu)
between 200 and 600 m within the 45-55ºN latitude band, again likely related to the
propagation of the Mediterranean overflow in the Atlantic Ocean. Conversely, it is 0.75 psu
fresher in the top layer north of 60ºN. The differences between GLOB16 and climatologies for
both fields are small below 1500 m depth.

**3.2 Volume and heat transports**
Transports, in particular the meridional overturning circulation (MOC), are frequently used to
evaluate the model performance. To provide an overview of the large-scale general circulation
of the GLOB16 model, we present the time-mean meridional overturning stream function of
the flow for a zonally averaged view. The MOC, displayed in depth space, is shown in Fig. 3
for the Atlantic and the Indo-Pacific basins as well as for the global domain. In GLOB16, the
Atlantic overturning (AMOC, Fig. 3a) reproduces the two overturning cells linked to the
formation of North Atlantic Deep Water (NADW) and Antarctic Bottom Water (AABW). It
consists of northward surface flow in the top 1000 m, sinking north of 45° (with ~6 Sv sinking
north of the Greenland Scotland Ridge), and a southward return flow mainly occurring
between depths of ~1000 and 3000 m. It reaches its maximum strength of ~20 Sv at a depth of
1000 m around 35ºN. The AABW cell fills the deep ocean below 3000 m, and reaches ~6 Sv.
The cross-equatorial transport is ~16.5 Sv.
Relevant measurements with respect to the mass transport in the Atlantic Ocean and the
associated heat transport are provided by the RAPID/MOCHA program (e.g., Cunningham et
al. 2007) that makes the net transport across 26.5°N available since spring 2004. Both models
are in very good agreement with the RAPID observations at 26.5ºN. The GLOB16 overturning
strength and variability, computed at that latitude for the simulated decade, is 20.1 ± 2.9 Sv,
which is stronger than, but reasonably consistent with the RAPID estimates of 17.0 ± 3.6 Sv
observed between April 2004 to December 2013 (McCarthy et al. 2015) (Table 2). The
GLOB16 and RAPID mean values for the 2009-2013 period are 19.3 ± 3.1 and 15.6 ± 3.2,
respectively (Table 1). In Fig. 3b, we compare the time series of the strength of the AMOC at
26.5ºN from the eddying model integration and the RAPID estimates. At that latitude,
GLOB16 simulation realistically reproduces the AMOC temporal variability on seasonal and
inter-annual time scales, although the simulated variability is lower than the observed. The
high-resolution model misrepresents the two events of low AMOC observed in 2009 and 2010,
when GLOB16 transport exhibits a clear, but much weaker than RAPID, decline. Time series
from the twin 1/4º simulation is also shown. The Atlantic overturning transport is generally
weaker in GLOB4, having a mean magnitude of 14.9 ± 2.6 Sv over the 10 simulated year,
~25% lower than the eddying model. GLOB4 underestimates RAPID values in the first
simulated years, closely follows RAPID from 2008, and does better capture the interannual
variability and the 2009-10 AMOC reductions. Stepanov et al. (under review) suggested that
the source of discrepancy between the two models in simulating the AMOC minima at 26.5ºN



might be related to the RAPID methodology used for the calculation, which does not fully take
into account the impact of the recirculation of the subtropical gyre on the mid-ocean transport.
Coarser resolution models, which cannot resolve processes near the western boundary, produce
weaker recirculation cell (e.g., Getzlaff et al. 2005, Roussenov et al. 2008, Zhang 2010).
Therefore, in GLOB4, a smaller impact of recirculation and eddies leads to a closer
correspondence between the model output and RAPID data. Table 1 shows that the good
agreement between GLOB16 and RAPID is true not only for the total AMOC transports, but
also for its components (the Florida Current, Ekman and the mid-ocean transports). Details on
the decomposition of the AMOC reproduced at 26.5ºN are given in Stepanov et al. (under
review).
The Indo-Pacific stream function with its intense equatorial upwelling is shown in Fig. 3c.
Apart from the uppermost layers, where Ekman transports dominate, the Indo-Pacific is filled
by the AABW cell that reaches its maximum values of ~18 Sv between 3000 and 4000 m
depth. As expected, the southward flow outcrops in the Northern Hemisphere consistently with
intermediate water formation and penetration of water from the circumpolar area near surface
and bottom, sandwiching a southward return flow at intermediate depths. The global MOC
(Fig. 3d) shows the northward flow in the upper ocean, ultimately reaching the North Atlantic,
the deep waters formed in the north (NADW and the diffusively formed Indian Deep Water
and Pacific Deep Water) that moves toward the Southern Ocean, where the directly wind-
driven circulation is represented by a strong Deacon cell that peaks to ~27 Sv at ~200 m depth
around 45ºS.
In the North Atlantic, the modelled overturning transport is associated with about 1 PW (1 PW
= $10^{15}$ W) of northward heat flux. The 5-year mean meridional heat transport (MHT) for the
Atlantic Ocean simulated by GLOB16 is presented in Fig. 4a; transports from GLOB4 and
observational estimates are shown for comparison. It is worth noting that the heat transport
magnitude and the location of its maximum are data dependent, although the latitudinal
variation is comparable among them. The variation with latitude of the GLOB16 transport
realistically follows observed profiles; its magnitude is positive at all latitudes, consistent with
heat being carried northward in both hemispheres of the Atlantic Ocean, and larger than
GLOB4 in most of the basin. GLOB16 generally underestimates the heat transport relative to
in situ measurements, as also seen in the COREII coarse-resolution models analysed by
Danabasoglu et al. (2014) and in the 1/10º climate model by Griffies et al. (2015). However,
our eddying-model MHT lies between implied transport estimates: in particular, it is generally
below the transport derived from Large and Yeager (2009), but it is always larger than
estimates by Trenberth and Fasullo (2008). The MHT maximum is found at ~22°N by Large
and Yeager (2009), and is more widely distributed between 20° and 30° N in the estimates of
Trenberth and Fasullo (2008). In GLOB16, the MHT reaches 1.1 PW at ~24°, where
observations by Lumpkin and Speer (2007) and Ganachaud and Wunsch (2003) are 1.24 ± 0.25
PW and 1.27 ± 0.15 PW, respectively. The distinct contributions from the overturning and the
gyre circulations to ocean heat transport are also computed (according to Johns et al. 2011) and
included in Fig. 4a. The overturning contribution dominates over a large latitude range. This is



particularly the case between the Equator and 25°N where the overturning component is within
one standard deviation of the mean total heat transport. Poleward, the MOC component drops,
while the gyre component increases explaining the large GLOB16 MHT north of 40ºN (in
agreement to the eddying climate model results by Griffies et al. 2015). The gyre transport
becomes comparable to the overturning contribution at ~45°N, and dominating the Atlantic
heat transport from 60°N. Apart from the North Atlantic subpolar gyre, the gyre contribution is
relevant between 10°S and the equator, where the gyre and overturning components contribute
about equally to the total heat transport. In GLOB4, the positive MHT slope between 45ºN and
55ºN indicated a large gain of heat. It is worth noting that this feature, present in many coarse
and eddy-permitting models (e.g. Danabasoglu et al. 2014, Grist et al. 2010), is absent in
GLOB16, likely due to a correct path of the simulated North Atlantic Current (Danabasoglu et
al. 2014, Treguier et al. 2012), as described in Sect. 3.6.
At 26.5°N, despite a stronger-than-observed AMOC magnitude, GLOB16 underestimates the
Atlantic heat transport estimates all through the 10-year RAPID record (2004-2013). Similar
behaviour can be seen in many model studies covering a large range of horizontal resolution
(e.g., Maltrud and McClean 2005, Mo and Yu 2012, Haines et al. 2013, Danabasoglu et al.
2014). The simulated MHT is lower by ~10% than mean RAPID value that equals 1.24 PW
(McCarthy et al. 2015), but the model output agrees, to a greater extent, with the most recent
RAPID estimates, which show a decrease of MHT since 2009: the 5-year means of $1.31 \pm 0.27$
PW for the pentad 2004-2008 drops by 15% to $1.14 \pm 0.08$ PW for the pentad 2009-2013. The
variation in time of the modelled and observed MHT at 26.5ºN is presented in Fig. 4b. Both
runs misrepresent the large summer fluxes in the first 2 years of integration. Afterwards,
GLOB16 matches very closely the RAPID magnitude and its variability from 2006 on. The
eddy-permitting GLOB4, instead, underestimates both the eddying configuration and the
RAPID record with a mean value and variability of $0.87 \pm 0.21$ PW.
**3.3 Volume transports through critical sections**
Although the two models do generally reproduce similar large-scale ocean circulation,
performing high-resolution simulations alters strength, shape and position of the main gyres
(Lévy et al. 2010), but especially results in a more accurate representation of narrow boundary
currents. To judge the level of agreement between the model velocity fields and the
observational data, we list, in Table 2, the time-mean volume transports through well-defined
critical straits and passages, evaluated from GLOB16 velocities averaged over the 10 years of
integrations, together with GLOB4 values, observation-based estimates and their sources for
each region. It is worth noting that the observational products are based on numbers of
assumptions and do not always cover the simulated decade.
The strengths of the GLOB16 transports agree well with observations, and are generally within
or very close to the limits of observed uncertainty. First, we consider the Drake Passage
transport as representative of the large-scale features of the Antarctic Circumpolar Current
(ACC). The zonal circumpolar transport ranges between about 112 Sv and 137 Sv, with a mean



value of 122.6 Sv, comparable to the recent observational estimate over the period 2007-2011
by Chidichimo et al. (2014) and close to the lower bound of the canonical ACC transport from
Cunningham et al. (2003). The time-series of the monthly averaged transport, in Fig. 5a, shows
a decline of ~10 Sv in the first 3 simulated years, then the drift becomes negligible. As shown
by Farneti et al. (2015), at coarser resolution, the mean transport is generally larger than
observational estimates. The increase in resolution largely improves the mean ACC transport,
which is ~20% stronger in GLOB4.
The total Indonesian throughflow (ITF) volume transport estimates from the 3-year INSTANT
Program corresponds to 15.0 Sv, varying from 10.7 to 18.7 Sv (Sprintall et al. 2009). The
mean ITF transport from GLOB16 (computed at 114ºE, between Indonesia and Australia) falls
within this range, but slightly overestimates the observed mean value. The GLOB16
contributions to the Pacific-to-Indian Ocean flow across Lombok, Ombai and Timor straits
follow within the range of minimum and maximum values from INSTANT (Sprintall et al.
2009, Gordon et al. 2010). Beside a weak decrease in the first years of simulation, the ITF has
no evident drift over time (Fig. 5b). In GLOB4, the total mean value is closer to observations,
but its decomposition is not: the Lombok Strait is closed and is likely compensated by a too
strong transport through the Ombai strait.
The flux across the Mozambique Channel simulated by both models follows within the broad
range of observed estimates. GLOB16 time series, in Fig. 5c, is characterized by a large
seasonal cycle and is free from any significant drift.
Comparing the strength of the modelled and observation-based volume transports through the
main Arctic Ocean gateways shows that GLOB16 calculations lie within the observed mean
values and within the uncertainty range of observations in these areas. The simulated Pacific
inflow across the Bering Strait of 1.1 Sv is consistent with observed values in both models,
overestimating the recent estimates by Woodgate et al. (2012) to a small degree. The large
transport at Bering Strait is common to other NEMO simulations, also at high-resolution (e.g.
Marzocchi et al. 2015). For the average outflow from the Arctic Ocean (computed across Fram
and Davis straits), the simulated 4.6 Sv are indistinguishable from observations, reproducing a
correct partitioning of the exports west and east of Greenland. 2.4 Sv flow southward across
the Fram Strait, compared with an observational estimates of 2 ± 2.7 Sv (Schauer et al. 2008),
and 2.2 Sv in the Davis Strait against estimates of 2.6 ± 1 Sv (Cuny et al 2005) and more recent
1.6 ± 0.5 Sv (Curry et al 2014). The seasonal cycles of the two transports are out of phase,
indicating that the fluxes out of the Arctic Ocean across those strait partially balance each other
(Fig. 5d). In contrast, GLOB4 reproduces a stronger transport through the Canadian
Archipelago, and underestimates the Fram Strait component.
The dense and cold overflows from the Nordic Seas supply the densest waters to NADW (e.g.
Eldevik et al. 2009) and have a fundamental impact on the circulation in the Irminger and
Labrador Seas, which are active sites of deep-water formation (e.g. Dickson et al. 2008). To
assess whether GLOB16 is capable to reproduce the strength of the overflow (here defined as
$\sigma_\theta > 27.8$ kg m$^{-3}$), the corresponding volume transport has been calculated both in the Denmark
Strait and in the Faroe Bank Channel. The mean transport appears to be consistent with



observations in the Denmark Strait, with a mean overflow transport of 2.7 Sv across the
Denmark Strait, which slightly underestimates the long-term observed transport of ~3 Sv
(Macrander et al. 2007, Jochumsen et al. 2012). There is no clear seasonal cycle, and no
discernible trend is detected for the whole period (Fig. 5e), as observed by Dickson et al.
(2008). The mean transport of dense water across the Faroe Bank Channel is 1.7 Sv with
absent trend (Fig. 5e), in well accordance with the observed values of ~2 Sv (Hansen and
Østerhus 2007). This consistency builds confidence that the dense water transport processes are
realistically simulated in GLOB16. At lower resolution, water-masses at the sill depth in the
Denmark Strait are too light compared with observations, resulting in a weak overflow in the
considered density class; while the Faroe Bank Channel overflow is too dense, with a
consequent large transport.


**3.4 Mixed layer depth**
Here we evaluate the winter mixed layer depth (MLD) in both hemispheres. MLDs are
computed using a density threshold of 0.03 kg m$^{-3}$ from the near-surface value. The two
models represent the mixed layer quite realistically, across the global domain, with similar
spatial distribution. Fig. 6 shows the GLOB16 MLD for March (September) in the Northern
(Southern) Hemisphere calculated for years 2009-2013, alongside the reconstructed
climatology of de Boyer Montégut et al. (2004) for the 1994-2002 period. In general, GLOB16
realistically reproduces the expected spatial patterns of the winter surface mixing, with good
correspondence between regions of shallow and deep mixed layers. The model reproduces
regions of shallow MLDs in the tropics. Locations of maxima are realistic both in the northern
and the southern hemispheres. In the North Atlantic, the sites of winter dense-water formation
are realistically located in the subpolar gyre, with the deepest mixing occurring in the Labrador
Sea, where it reaches over 2000 m  (Fig. 7). In the Nordic Seas, the winter mixing is strong
along the path of transformation of Atlantic water in the Norwegian Sea and convective site are
reproduced south of Svalbard and in the Iceland Sea with MLDs down to 400 and 1000 m
depth, respectively. In the Northern Hemisphere, both runs reproduce mixed layer maxima
deeper than observed estimates, as generally seen in NEMO calculations at different
resolutions (e.g. Megann et al. 2014, Marzocchi et al. 2015). In GLOB4, the winter mixing in
the Nordic Seas is comparable to GLOB16 results, while in the Labrador Sea is shallower than
GLOB16 (Fig. 7), but covering a much wider area (not shown). In the austral hemisphere, the
deepest winter mixed layer corresponds to the near-zonal bands of deep turbulent mixing along
the path of the ACC, where the mixed layer deepens in many instances (Sallée et al. 2010).
Maximum values of ~800 m are found in the Pacific basin, not exactly collocated with the
observed one (Fig. 6). Both models have a significant deeper mixed layer in regions of AABW
formation, associated with densification of the water masses over the Antarctic continental
shelf, a result similarly shown in a recent COREII study assessing 15 ocean-sea ice models
(Downes et al. 2015). The mixed layer reaches depths of 500 m and 400 m over the Ross Sea
and the Weddell Sea continental shelves, respectively. GLOB4 mixed layer is deeper in the



Southern ocean, reaching to over 4000 m in many instances in the first years of integration
(Fig. 7).

**3.5 Sea ice**
Formation and melting of sea ice strongly affect the ocean dynamics both locally in polar
regions and in the global ocean, through the contribution of high-latitude processes in deep
water production. Here we present sea ice properties and their variability for both hemispheres
as simulated by the numerical experiments in comparison with satellite observations. The mean
fields are computed over the period 2009-2013, excluding the first 5 years of integration in
which the sea ice model is far from the equilibrium. Sea ice extent is defined as the area of the
ocean with an ice concentration of at least 10%.
In Fig. 8a, the mean seasonal cycle of sea ice extent reproduced by GLOB16 is compared with
products from passive microwave satellites SSM/I processed at the National Snow and Ice
Data Center (NSIDC, Cavalieri et al. 1996) for both the north and south polar regions. In the
Arctic Ocean, the simulated mean extent of $9.5 \times 10^6$ km$^2$ and the amplitude of the seasonal
cycle of $10.3 \times 10^6$ km$^2$ are, to a great extent, in good agreement with the observations
($10.8 \times 10^6$ km$^2$ and $10.7 \times 10^6$ km$^2$, respectively). Although the mean sea ice extent is smaller
than the satellite estimates by ~10% year-round, the GLOB16 results are largely improved in
the end of the run, when the sea ice extent seasonal cycle approaches closely the satellite
estimates for both minima and maxima. These results suggest that GLOB16 is able to well
represent the sea ice thermodynamics processes after 10 years of integrations.
Figure 8b presents the seasonal cycle of Arctic sea ice volume as simulated in GLOB16 and
estimated by the data-assimilative model PIOMAS (Pan-Arctic Ice Ocean Modeling and
Assimilation System), which compares well with ICESat and CryoSat2 estimates and can be
reasonably considered a proxy for reality (Schweiger et al. 2011). From 2009 on, the GLOB16
sea ice volume ($14.4 \times 10^3$ km$^3$) matches very closely PIOMAS values ($14.5 \times 10^3$ km$^3$), even if
the modelled Arctic sea ice is slightly too thick (thin) during the melting (growing) season. The
maximum sea ice volume in GLOB16 is anyway overestimated in winter 2011 and 2012 (not
shown), following an increase of thickness due to sea ice drift and then mechanical processes.
Overall, the sea ice drift in the Arctic Ocean is similar to what is expected. The transpolar drift
and the Beaufort gyre circulation patterns are realistically simulated, but ice velocities are
generally too high. Nevertheless, the ice area flux of $74.9 \times 10^3$ km$^2$ month$^{-1}$ across Fram Strait
in the simulated decade matches very well to estimates of 75.8 based on using Advanced
Synthetic Aperture Radar (ASAR) images and passive microwave measurements (Kloster and
Sandven 2011), probably compensated by lower thickness (Fig 8c). The Arctic sea ice extent
and volume and their variability in time simulated by GLOB4 almost coincide with GLOB16
output, having mean sea ice extent of 9.3 (10.9)$\times 10^6$ km$^2$ and mean volume of 14.3 (7.1)$\times 10^3$
km$^3$ in the northern (southern) hemisphere. GLOB4 underestimates the observed ice area
export out of the Arctic Ocean through the Fram strait by ~13%, with a mean value of 66.1
$\times 10^3$ km$^2$ month$^{-1}$.



In the Southern Hemisphere, sea ice extent simulated by the two models is again consistent
with observations, but GLOB16 (GLOB4) undervalues the total sea ice extent by 1.6 (1.8)
$\times 10^6$ km$^2$. The low maximum in September accelerates the melting process and results in a
larger minimum in February (Fig. 8a). At present, no published long-term record of sea ice
volume are available for the Southern Hemisphere, making a formal validation of the model
skills in simulating sea ice volumes in that region unachievable. We consider recent ICESat
laser altimeter observations covering the period 2003-2008 (Kurts and Markus 2012) for a
qualitative comparison with model outputs, although uncertainties are still high (Kern and
Spreen 2015). Due to the lower minimum sea ice concentration, both models also likely
underestimate sea ice thickness and volume in the austral summer, with a possible feedback on
the winter sea ice properties. GLOB16 total volume of ice varies substantially over the annual
cycle, with a growth of ~14000 km$^3$ in fall larger than the ~8800 km$^3$ by ICESat (Fig. 8b).
The sea ice edge and the ice geographical distribution are generally well simulated in
GLOB16, particularly in winter. Comparison between the simulated fields of sea ice
concentration and the satellite-based estimates averaged over 2009-2013 shows that the
GLOB16 sea ice distribution in the end of the growing seasons is realistic in both hemispheres
(Fig. 9a,b and 10c,d), although the model simulates a much uniform sea ice concentration
around Antarctica (Fig. 10c,d). Summer minima are well reproduced in terms of ice edge, but
the regional concentration shows differences from the observations (Fig. 9c,d, and 10a,b). In
the Arctic Ocean, the GLOB16 reproduces the maximum ice concentration close to the
Canadian archipelago, but the spatial structure is misrepresented over a large area, with too low
sea ice concentration in the eastern-central sector. This is likely to be caused by the SST
restoring and to a generally too warm Atlantic Water inflow.
The spatial distribution of the sea ice in March is correctly reproduced in the Southern Ocean,
with the highest value in the Ross Sea and close to the Antarctic Peninsula in the Weddell Sea,
where the area of maximum concentration is anyway smaller than the observed one. The too
low ice concentration in the austral summer is constantly simulated from the beginning of the
run, and might be related to a too small sea ice concentration used to initialise the simulation.

**3.6 Mesoscale variability**
To assess the dynamical capacities of the GLOB16 configuration and to evaluate the gain in
representing mesoscale variability due to the higher resolution, Fig. 11 show maps of the sea
surface height (SSH) variability, represented by the standard deviation plots, from the eddying
ocean compared with the eddy-permitting one and altimetry estimates from AVISO product.
The spatial structure and intensity of the SSH variability can be used as indicator of strengths
and deficiencies of the mean flow. Both models reproduce the major circulation features
estimated from satellite measurements. Large values are collocated with the major current
systems associated with the Kuroshio Current, the Gulf Stream, the Loop Current in the Gulf
of Mexico, the strong equatorial current system and, in the southern ocean, the Eastern
Australian and the Leeuwin currents, the Brazil and Malvinas current system, the Agulhas



Current and the Antarctic Circumpolar Current. Although GLOB4 does a credible work of
reproducing the general observed spatial pattern, it simulates vast areas of low SSH variability
in the ocean interior, which indicates weaker flow instabilities and fewer meanders. GLOB16
shows additional instabilities in the upper ocean with a spatial structure richer in mesoscale
features that cover most of the ocean surface, and is more consistent to the observational
estimates.
Examination of individual regions can highlight the improvements in GLOB16. In the
Northern Hemisphere, the western boundary currents and their extensions are more sharply
reproduced at higher resolution. For example, even if the separation point of the Gulf Stream is
not largely modified (at ~37°N), its path and areal extent differ largely between configurations.
The GLOB16 current turns northwestward around the Grand Banks, instead continuing
eastward across the Atlantic (as in GLOB4). Further offshore, the current separates into a
southern branch heading toward the Azores Islands and a second branch flowing towards
Newfoundland. This feature is not correctly reproduced in the eddy-permitting case, as in many
coarser resolution models, leading to a cold and fresh bias in the northwestern subpolar gyre.
The separation of the Kuroshio Current occurs at about the same latitude (~36°N) in both
models, but the high variability region of the Kuroshio extension extends out to 180ºE in
GLOB16 in close agreement with data, while only reaches to 160ºE in GLOB4.
Some characteristic aspects of the global current systems are still misrepresented, also in the
eddying run. The performance of GLOB16 in reproducing the observed magnitude of the SSH
variability is a clear weakness. In many locations in the Southern Ocean, the GLOB16 map
shows a wider and more homogeneous distribution of oceanic eddies, but mesoscale turbulence
tends to be organized into a large numbers of small and relatively weak patches. The local
variability in the 1/16° simulation becomes comparable to or lower than that in the 1/4°
simulation and the altimeter map. This is pronounced within the main body of the ACC where
local maxima have not substantially and positively increased with resolution. In the Agulhas
region, the model shows a band of high variability along the paths of the Mozambique Current,
the East Madagascar Current, and the Agulhas retroflection, but the modelled SSH variability
is again much less than the observed one. In the Brazil Malvinas convergence region the SSH
variability presents a local minimum at about 55°W, 42°S but does only partially resemble the
observed *C*-shape. Modelled magnitude departs significantly from observations also in the East
Australian Current.
SSH variance distribution shows strong qualitative similarities to the EKE for the near surface
(not shown). In Fig. 12a, we shows the surface EKE, zonally averaged, as calculated from the
two simulations and derived from the OSCAR data set (Ocean Surface Current Analyses Real-
time, Bonjean and Lagerloef 2002). OSCAR provides estimates of near-surface ocean currents
on a 1/3° grid with a 5 day resolution, combining scatterometer and altimeter data.
Quantitatively the models differ significantly from the observations, GLOB16 being the
closest. However, both models reproduce higher levels of EKE concentrated at the latitude of
the major current system, at the Equator, about 40° N in the Northern Hemisphere and linked
to the ACC and the main western boundary currents in the Southern Ocean. The zonal-



averaged EKE profiles emphasize that, despite the local defects, the GLOB16 surface levels of
energy exceeds GLOB4 everywhere, except in the equatorial band where the westward
extension of the Pacific currents is less pronounced. For the higher resolution model, the
surface EKE increases by ~20% relative to GLOB4. Since the two models are forced by
identical atmospheric fields, the increase in EKE with resolution arises primarily from
increased baroclinic and barotropic instability of the mean flow in the high-resolution model,
which tends to generate more meanders and eddies. It has been shown that higher level of near
surface EKE closer to the one derived from OSCAR can be obtained by assimilating in-situ
and altimeter data in a set of eddy-permitting ORCA025 configurations (Masina et al., 2015).
In particular, the assimilation of sea level anomaly has been proven to be effective in
introducing mesoscale variability (Storto et al., 2015) underestimated by an eddy-permitting
configuration similar to the one used in this work. Our results suggest that the increased
resolution of GLOB16 is also able to partially recover part of the observed variability.
However, GLOB16 value represents only ~60% of the surface EKE estimated from OSCAR.
The kinetic energy of the mean flow (MKE) at surface is similar between the models. It
increases by 5% in the 1/16° simulation, reaching 94% of the observed MKE (Fig. 12b).

**4. CONCLUSIONS**
We have introduced a new global eddying-ocean model configuration, GLOB16, developed at
CMCC, and presented an overview from an 11-year simulation. GLOB16 is an implementation
of version 3.4 of the NEMO model, with horizontal resolution of at least 1/16° everywhere and
98 vertical levels, together with the LIM2 sea ice model on the same grid.
Overall, the model results are quite satisfactory when compared to observations and the gain
due to increased resolution is evident when compared to a coarser-resolution version of the
model. Analysis of the model zonally-averaged temperature and salinity, MLD, overturning
circulation and associated northward heat transport, lead us to conclude that the model average
state is realistic, and that the model realistically represents the variability in the upper ocean
and at intermediate depths. GLOB16 model configuration showed good skill in simulating
exchanges of mass between ocean basins and through key passages. The contributions from the
individual straits in the exports from the Arctic Ocean are within the uncertainties of the
observational estimates. The seasonal cycles of total ice area and volume are close to satellite
observations and the sea ice extent distribution is very well reproduced in both hemispheres,
although sea ice concentration and thickness can be further improved together with sea ice
drift. The model is able to hindcast the position and strength of the surface circulation.
Comparisons between the SSH variability from the model and from gridded observations
indicate that the model variability is acceptable, with local maxima and minima in the same
locations as observations. Extension and separation of western boundary currents are better
resolved compared to the eddy-permitting run. However, a clear weakness of the GLOB16
model is its ability in reaching the observed magnitude of the SSH variability, especially in the
Southern Ocean. This behaviour is most likely related to the coefficients chosen for vertical



and lateral eddy diffusivity and viscosity, and detailed numerical studies are planned to improve
these aspects. It is also possible that the relatively coarse resolution (~0.75°) of the ERA-
Interim wind forcing may play a partial role on this underestimation, and whether higher-
resolution atmospheric products can overcome this feature is to be investigated.
In spite of its shortcomings, we think that GLOB16 represents a significant modelling
improvement over the previous configurations of the CMCC global ocean/sea ice models at
coarser resolutions. As our first step in exploring the behaviour and fidelity of eddying global
models, this simulation sets the necessary groundwork for further, more detailed studies. To
potentially ameliorate the model realism, we plan, in the near future, to improve physical
parameterizations and include physics upgrades either available or under development in
NEMO, such as the full non-linear free surface physics, Langmuir turbulence scheme, vertical
mixing parameterizations. We expect that these developments will help address some of the
shortcomings identified in this study.
The next phase will be to couple GLOB16 to an ocean/sea ice data assimilation system, similar
to that described by Storto et al. (2015). Subsequent to that activity, GLOB16 will constitute
the base of a global eddying analysis and short-term forecast system, intended to provide
boundary conditions for downscaling and forecasting nested models in the world oceans.

**Code availability**
The NEMO model is freely available under the CeCILL public licence. After registration on
the NEMO website (http://www.nemo-ocean.eu/), users can access the code (via Subversion,
http://subversion.apache.org/) and run the model, following the procedure described in the
"NEMO Quick Start Guide". The revision number of the code used for this study is 4510. The
CMCC NEMOv3.4 code includes some additional modifications, applied to the base code. In
particular, we modified the North Pole folding condition, introducing a more sophisticated
optimization of the north fold algorithm (Epicoco et al. 2014), which leads to an extra increase
in model performances (up to 20% time-reduction on the used architecture) without altering
any physical process. The algorithm is now available in NEMO version 3.6. Interested readers
can contact the authors for more information on the CMCC NEMOv3.4 code.

**Acknowledgements**
The financial support of the Italian Ministry of Education, University and Research, and Ministry for
Environment, Land and Sea through the project GEMINA is gratefully acknowledged. We also acknowledge
PRACE for awarding us the project ENSemble-based approach for global OCEAN forecasting (ENS4OCEAN)
and providing access to resource on MareNostrum based in Spain at Barcelona Supercomputing Center. The
numerical results used here are available under request at CMCC. The RAPID data have been loaded from the
following web pages: http://www.rapid.ac.uk/rapidmoc and http://www.rsmas.miami.edu/users/mocha. The EN3
subsurface ocean temperature and salinity data were collected, quality-controlled and distributed by the U.K. Met
Office Hadley Centre. The Aviso altimeter products were produced by Ssalto/Duacs and distributed by Aviso,
with support from CNES.



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

G. Yeager: An assessment of Antarctic Circumpolar Current and Southern Ocean meridional
overturning circulation during 1958–2007 in a suite of interannual CORE-II simulations.
Ocean Model., 93, 84-120, 2015.
Fichefet, T., and M.A. Morales Maqueda: Sensitivity of a global sea ice model to the treatment
of ice thermodynamics and dynamics. J. Geophys. Res., 102, 12609–12646, 1997.
Fretwell, P., H. D. Pritchard, D. G. Vaughan, J. L. Bamber, N. E. Barrand, R. Bell, C. Bianchi,
R. G. Bingham, D. D. Blankenship, G. Casassa, G. Catania, D. Callens, H. Conway, A. J.
Cook, H. F. J. Corr, D. Damaske, V. Damm, F. Ferraccioli, R. Forsberg, S. Fujita, P. Gogineni,
J. A. Griggs, R. C. A. Hindmarsh, P. Holmlund, J. W. Holt, R. W. Jacobel, A. Jenkins, W.
Jokat, T. Jordan, E. C. King, J. Kohler, W. Krabill, M. Riger-Kusk, K. A. Langley, G.
Leitchenkov, C. Leuschen, B. P. Luyendyk, K. Matsuoka, Y. Nogi, O. A. Nost, S. V. Popov, E.
Rignot, D. M. Rippin, A. Riviera, J. Roberts, N. Ross, M. J. Siegert, A. M. Smith, D.
Steinhage, M. Studinger, B. Sun, B. K. Tinto, B. C. Welch, D. A. Young, C. Xiangbin, and A.
Zirizzotti: Bedmap2: improved ice bed, surface and thickness datasets for Antarctica. The
Cryosphere 7, 375-393, 2013.
Ganachaud, A., and C. Wunsch: Large-scale ocean heat and freshwater transport during the
World Ocean Circulation Experiment. J. Climate, 16, 696–705, 2003.
Getzlaff, J., C. Böning, C. Eden, and A. Biastoch: Signal propagation related to the North
Atlantic overturning. Geophys. Res. Lett., 32 (9), L09, 602, 2005.
Good, S.A., M.J. Martin, and N.A. Rayner: En4: quality controlled ocean temperature and
salinity profiles and monthly objective analyses with uncertainty estimates. Journal of
Geophysical Research: Oceans, 118, 6704–6716, 2013.
Gordon, A. L., J. Sprintall, H. M. Van Aken, D. Susanto, S. Wijffels, R. Molcard, A. Ffield,
W. Pranowo, and S. Wirasantosa: The Indonesian throughflow during 2004-
2006 as observed by the INSTANT program. Dynamics of Atmospheres and Ocean, 50, Issue
2, Pages 115-128, 2010.



Griffies, S. M., M. Winton, W. G. Anderson, R. Benson, T. L. Delworth, C. O. Dufour, J. P.
Dunne, P. Goddard, A. K. Morrison, A. T. Wittenberg, J. Yin, and R. Zhang: Impacts on ocean
heat from transient mesoscale eddies in a hierarchy of climate models. J. Climate, 28(3), 2015.

Grist, J. P., S.A. Josey, R. Marsh, S.A. Good, A.C. Coward, B.A. de Cuevas, S.G. Alderson, A.
New, L. Adrian, and G. Madec: The roles of surface heat flux and ocean heat transport
convergence in determining Atlantic Ocean temperature variability. Ocean Dynam., 60, (4),
857    771-790, 2010.

Haines, K., V.N. Stepanov, M. Valdivieso, and H. Zuo: Atlantic meridional heat transports in
two ocean reanalyses evaluated against the RAPID array. Geophys. Res. Lett., 40, 343–348,
861    2013.

Hallberg, R.: Using a resolution function to regulate parameterizations of oceanic mesoscale
eddy effects. Ocean Model., 72, 92–103, 2013.

Hansen, B., and S. Østerhus: Faroe Bank Channel overflow 1995-2005. Progr. Oceanogr., 75,
817–856, 2007.

Hunke, E.C., and J.K. Dukowicz: An elastic–viscous–plastic model for sea ice dynamics an
elastic–viscous–plastic model for sea ice dynamics an elastic–viscous– plastic model for sea
ice dynamics. J. Phys. Oceanogr., 27, 1849–1867, 1997.

IOC, IHO and BODC: Centenary Edition of the GEBCO Digital Atlas, published on CD-ROM
on behalf of the Intergovernmental Oceanographic Commission and the International
Hydrographic Organization as part of the General Bathymetric Chart of the Oceans, British
Oceanographic Data Centre, Liverpool, U. K., 2003.

Ingleby, B., and M. Huddleston: Quality control of ocean temperature and salinity profiles -
Historical and real-time data. J. Marine Syst., 65, 158–175, 2007.

Iovino, D., A. Storto, S. Masina, A. Cipollone, and V. Stepanov: GLOB16, the CMCC global
mesoscale-eddying ocean. Research Papers Issue RP0247, December 2014.

Jochumsen, K., D. Quadfasel, H. Valdimarsson, and S. Jónsson: Variability of the Denmark
Strait overflow: Moored time series from 1996–2011, J. Geophys. Res., 117, C12003, 2012.

Johns, W. E., M. O. Baringer, L. M. Beal, S. A. Cunningham, T. Kanzow, H. L. Bryden, J. J.
M. Hirschi, J. Marotzke, C. S. Meinen, B. Shaw, and R. Curry: Continuous, Array-Based
Estimates of Atlantic Ocean Heat Transport at 26.5°N. J. Climate, 24, 2429–2449, 2011.

Kern, S., and G. Spreen: Uncertainties in Antarctic sea-ice thickness retrieval from ICESat.
Annals of Glaciology, vol. 56, issue 69, pp. 107-119, 2015.

Kloster K., and S. Sandven: Ice motion and ice area flux in the Fram Strait at 79N using ASAR
and passive microwave for Feb. 2004 – Jul. 2010, Tech. Rep. 322, Nansen Environmental and
Remote Sensing Center, 2011.



Kurtz, N.T., and T. Markus: Satellite observations of Antarctic sea ice thickness and volume. J.
Geophys. Res. 117, C08025, 2012.
Large, W., and S. Yeager: Diurnal to decadal global forcing for ocean sea ice models: the data
set and fluxes climatologies, Rep. NCAR/TN-460+STR, National Center for Atmospheric
Research, Boulder, Colorado, 2004.
Large, W., and S. Yeager: The global climatology of an interannually varying air-sea flux data
set. Clim. Dynam., 33, 341 – 364, 2009.
Le Traon, P.-Y., D. Antoine, A. Bentamy, H. Bonekamp, L.A. Breivik, B. Chapron, G. Corlett,
G. Dibarboure, P. DiGiacomo, C. Donlon, Y. Faugère, J. Font, F. GirardArdhuin, F. Gohin,
J.A. Johannessen, M. Kamachi, G. Lagerloef, J. Lambin, G. Larnicol, P. Le Borgne, E.
Leuliette, E. Lindstrom, M.J. Martin, E. Maturi, L. Miller, L. Mingsen, R. Morrow, N. Reul,
M.H. Rio, H. Roquet, R. Santoleri & J. Wilkin: Use of satellite observations for operational
oceanography: recent achievements and future prospects. Journal of Operational
Oceanography, 8:sup1, s12-s27, 2015.
Lévy, M., P. Klein, A.-M. Tréguier, D. Iovino, G. Madec, S. Masson, and K. Takahashi:
Modifications of gyre circulation by sub-mesoscale physics. Ocean Model., 34, 1-15, 2010.
Locarnini, R.A., A.V. Mishonov, J.I. Antonov, T.P. Boyer, H.E. Garcia, O.K. Baranova, M.M.
Zweng, C.R. Paver, J.R. Reagan, D.R. Johnson, M. Hamilton, and D. Seidov: World Ocean
Atlas 2013, Volume 1: Temperature. S. Levitus, Ed., A. Mishonov Technical Ed.; NOAA
Atlas NESDIS 73, 40 pp., 2013.
Lumpkin, R., and K. Speer: Global ocean meridional overturning. J. Phys. Oceanogr., 37,
2550–2562, 2007.
Macrander, A., R.H. Käse, U. Send, H. Valdimarsson, and S. Jónsson: Spatial and temporal
structure of the Denmark Strait Overflow revealed by acoustic observations. Ocean. Dyn, 57,
75–89, 2007.
Madec, G. and M. Imbard: A global ocean mesh to overcome the North Pole singularity, Clim.
Dynam., 12, 381–388, 1996.
Madec, G. and the NEMO team: Nemo ocean engine - version 3.4. Technical Report ISSN No
1288–1619, Pôle de modélisation de l'Institut Pierre-Simon Laplace No 27, 2012.
Maltrud, M. E., and J. L. McClean: An eddy resolving global 1/10° ocean simulation, Ocean
Model., 8, 31–54, 2005.
Marzocchi, A., J. J.-M. Hirschi, N. P. Holliday, S. A. Cunningham, A. T. Blaker, and A. C.
Coward: The North Atlantic subpolar circulation in an eddy-resolving global ocean model. J.
Marine Syst.,142. 126-143, 2015.
McCarthy, G.D., D.A. Smeed, W.E. Johns, E. Frajka-Williams, B.I. Moat, D. Rayner, M.O.
Baringer, C.S. Meinen, J. Collins, and H.L. Bryden: Measuring the Atlantic Meridional
Overturning Circulation at 26°N. Prog. Oceanog., 130, 91-111, 2015.




Megann, A., D. Storkey, Y. Aksenov, S. Alderson, D. Calvert, T. Graham, P. Hyder, J.
Siddorn, B. Sinha: GO5.0: the joint NERC-Met Office NEMO global ocean model for use in
coupled and forced applications. Geosci. Model Dev. 7, 1069–1092, 2014.

Metzger, E.J., O.M. Smedstad, P.G. Thoppil, H.E. Hurlburt, J.A. Cummings, A.J. Wallcraft,
L. Zamudio, D.S. Franklin, P.G. Posey, M.W. Phelps, P.J. Hogan, F.L. Bub, and C.J. DeHaan:
US Navy operational global ocean and Arctic ice prediction systems. Oceanography 27(3):32–
955 43, 2014.


Mo, H.-E., and Y.-Q. Yu: Simulation of volume and heat transport along 26.5°N in the
Atlantic. Atmos. Oceanic Sci. Lett., 5, 373–378, 2012.

Morrow, R., and P.Y. Le Traon: Recent advances in observing mesoscale ocean dynamics with
satellite altimetry. Advances in Space Research, 50(8), 1062–1076, 2012.

Nurser, A.J.G., and S. Bacon: The Rossby radius in the Arctic Ocean. Ocean Science, 10, 967–
964 975, 2014.


Oke, P.R., D.A. Griffin, A. Schiller, R.J. Matear, R. Fiedler, J. Mansbridge, A. Lenton, M.
Cahill, M.A. Chamberlain, and K. Ridgway: Evaluation of a near-global eddy-resolving ocean
model, Geosci. Model Dev., 6, 591-615, 2013.

Reynolds, R.W., T.M. Smith, C. Liu, D.B. Chelton, K.S. Casey, and M. G. Schlax. Daily high-
resolution-blended analyses for sea surface temperature. J. Climate, 20, 5473–5496, 2007.

Roullet, G. and G. Madec: salt conservation, free surface, and varying levels: a new
formulation for ocean general circulation models, J. Geophys. Res., 105, 23927–23942, 2000.

Roussenov, V., R. Williams, C. Hughes, and R. Bingham: Boundary wave communication of
bottom pressure and overturning changes for the North Atlantic. J. Geophys. Res, 113 (C8),
C08042, 2008.

Smith, R.D., M.E. Maltrud, F. Bryan, and M.W. Hecht: Numerical simulation of the North
Atlantic Ocean at 1/10. J. Phys. Oceanogr., 30, 1532–1561, 2000.

Schweiger, A., R. Lindsay, J. Zhang, M. Steele, and H. Stern: Uncertainty in modeled arctic
sea ice volume. J. Geophys. Res., 116, C00D06, 2011.

Sprintall, J., S. E. Wijffels, R. Molcard, and I. Jaya: Direct estimates of the Indonesian
Throughflow entering the Indian Ocean: 2004–2006, J. Geophys. Res., 114, C07001, 2009.

Stepanov, V. N., D. Iovino, S. Masina, A. Storto, and A. Cipollone: Methods of calculation of
the Atlantic meridional heat and volume transports from ocean models at 26.5°N. *Under*
*review to Journal of Geophysical Research: Oceans.*

Storto A., S. Masina and A. Navarra: Evaluation of the CMCC eddy-permitting global ocean
physical reanalysis system (C-GLORS, 1982-2012) and its assimilation components. Quart. J.
R. Meteorol. Soc., doi: 10.1002/qj.2673, 2015.



Treguier, A.M., J. Deshayes, C. Lique, R. Dussin, J.M. Molines: Eddy contributions to the
meridional transport of salt in the North Atlantic. J. Geophys. Res: Oceans, 117 (C5), 2012.
Trenberth, K. E., and J. T. Fasullo: An observational estimate of inferred ocean energy
divergence. J. Climate, 38, 984–999, 2008.
Woodgate, R. A., T. J. Weingartner, and R. Lindsay: Observed increases in Bering Strait
oceanic fluxes from the Pacific to the Arctic from 2001 to 2011 and their impacts on the Arctic
Ocean water column, Geophys. Res. Lett., 39, L24603, 2012.
Zalesak, S. T.: Fully multidimensional flux corrected transport for fluids. J. Comput. Phys., 31,
335–362, 1979.
Zhang, R.: Latitudinal dependence of Atlantic meridional overturning circulation (AMOC)
variations. Geophys. Res. Lett., 37, L16703, 2010.
Zweng, M.M., J.R. Reagan, J.I. Antonov, R.A. Locarnini, A.V. Mishonov, T.P. Boyer, H.E.
Garcia, O.K. Baranova, D.R. Johnson, D. Seidov, and M.M. Biddle. World Ocean Atlas 2013,
Volume 2: Salinity. S. Levitus, Ed., A. Mishonov Technical Ed.; NOAA Atlas NESDIS 74, 39
pp., 2013.



Table 1. AMOC and its constituents with standard deviations, averaged within the 2009-2013 period as obtained
from RAPID observations and the two models at 26.5ºN. The modelled Gulf Stream transports include both the
Florida current and Western boundary current contributions.

|  | **RAPID** | **GLOB16** | **GLOB4** |
|---|---|---|---|
| AMOC | 15.6 ± 3.2 | 19.3 ± 3.1 | 14.3 ± 2.7 |
| Ekman | 3.3 ± 2.3 | 2.7 ± 2.4 | 2.7 ± 2.3 |
| Gulf stream | 31.2 ± 2.3 | 34.9 ± 2.7 | 32.2 ± 2.1 |
| Upper Mid-Ocean | -18.9 ± 2.8 | -19.8 ± 2.0 | -21.3 ± 1.6 |
| Throughflow | 0 | -1.6 ± 0.5 | -0.8 ± 0.5 |

Table 2. Volume transports (in Sv) through key sections, simulated values averaged in the 2004-2013 period and
observed mean values with their standard deviations (when available). Positive values correspond to northward
and eastward flows.

|  | GLOB16 | OBSERVED | | GLOB4 |
|---|---|---|---|---|
| max AMOC at 26.5°N | 20.1 ± 2.9 | 17 ± 3.6 | McCarthy et al. 2015 | 14.9 ± 2.6 |
| Drake Passage | 122.6 ± 5.7 | 136.7 ± 6.9 | Cunningham et al. 2003 | 149.5 ± 9.5 |
|  |  | 127.7 ± 8.1 | Chidichimo et al. 2014 |  |
| ITF (total at 114°E) | -18.1 ± 2.5 | -15 ± 4 | Sprintall et al. 2009 | -16.1 ± 2.8 |
| Lombok Strait | -2.2 ± 1.9 | -1.8 to -3.2 | Sprintall et al. 2009 | -------- |
|  |  | -2.6 | Gordon et al. 2010 |  |
| Ombai Strait | -4.7 ± 2.2 | -2.7 to -5.0 | Sprintall et al. 2009 | -5.7 ± 1.4 |
|  |  | -4.9 | Gordon et al. 2010 |  |
| Timor Passage | -6.8 ± 1.8 | -6.2 to -10.5 | Sprintall et al. 2009 | -7.2 ± 1.6 |
|  |  | -7.5 | Gordon et al. 2010 |  |
| Mozambique Channel | -23.4 ± 5.4 | -29.1 | DiMarco et al. 2002 | -20.8 ± 5.8 |
|  |  | -16.7 | van der Werf et al. 2010 |  |
| Bering Strait | 1.1 ± 0.5 | 0.8 ± 0.2 | Woodgate et al. 2012 | 1.1 ± 0.5 |
| Fram Strait | -2.4 ± 1.0 | -2.0 ± 2.7 | Schauer et al. 2008 | -1.5 ± 1.2 |
|  |  | -2.3 ± 4.3 | Curry et al. 2011 |  |
| Davis Strait | -2.2 ± 0.5 | -2.6 ± 1.0 | Cuny et al. 2005 | -3.4 ± 0.9 |
|  |  | -1.6 ± 0.5 | Curry et al. 2014 |  |
| Denmark Strait overflow | -2.7 ± 0.4 | -3.4 ± 1.4 | Jochumsen et al. 2012 | -1.4 ± 0.3 |
| FBC overflow | -1.7 ± 0.2 | -1.9 ± 0.3 | Hansen and Østerhus 2007 | -2.5 ± 0.3 |




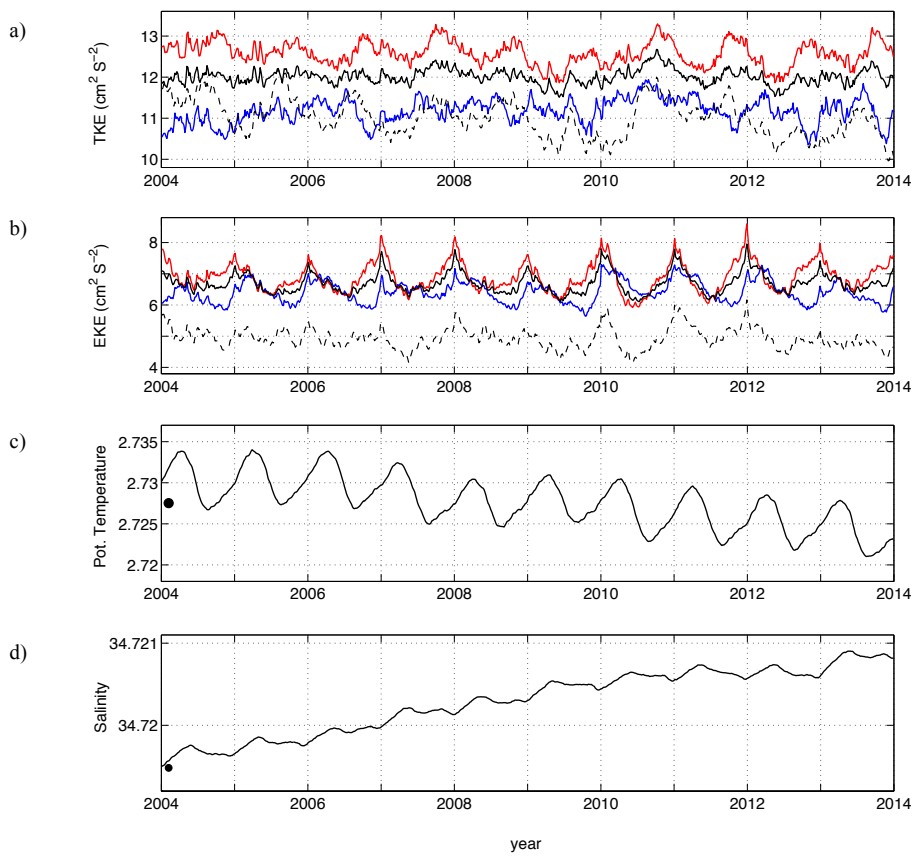

Fig. 1. Time variations of volume-averaged (a) TKE (in cm$^2$ s$^{-2}$), where the black line represents the global basin-
mean value and the red (blue) the contribution of the Southern (Northern) Hemisphere in GLOB16. Thin-dashed
line represents the basin-mean TKE in GLOB4. (b) As (a) but for EKE (in cm$^2$ s$^{-2}$). (c) Potential temperature in
°C, and (d) salinity in psu. Black circles indicate temperature and salinity initial values.

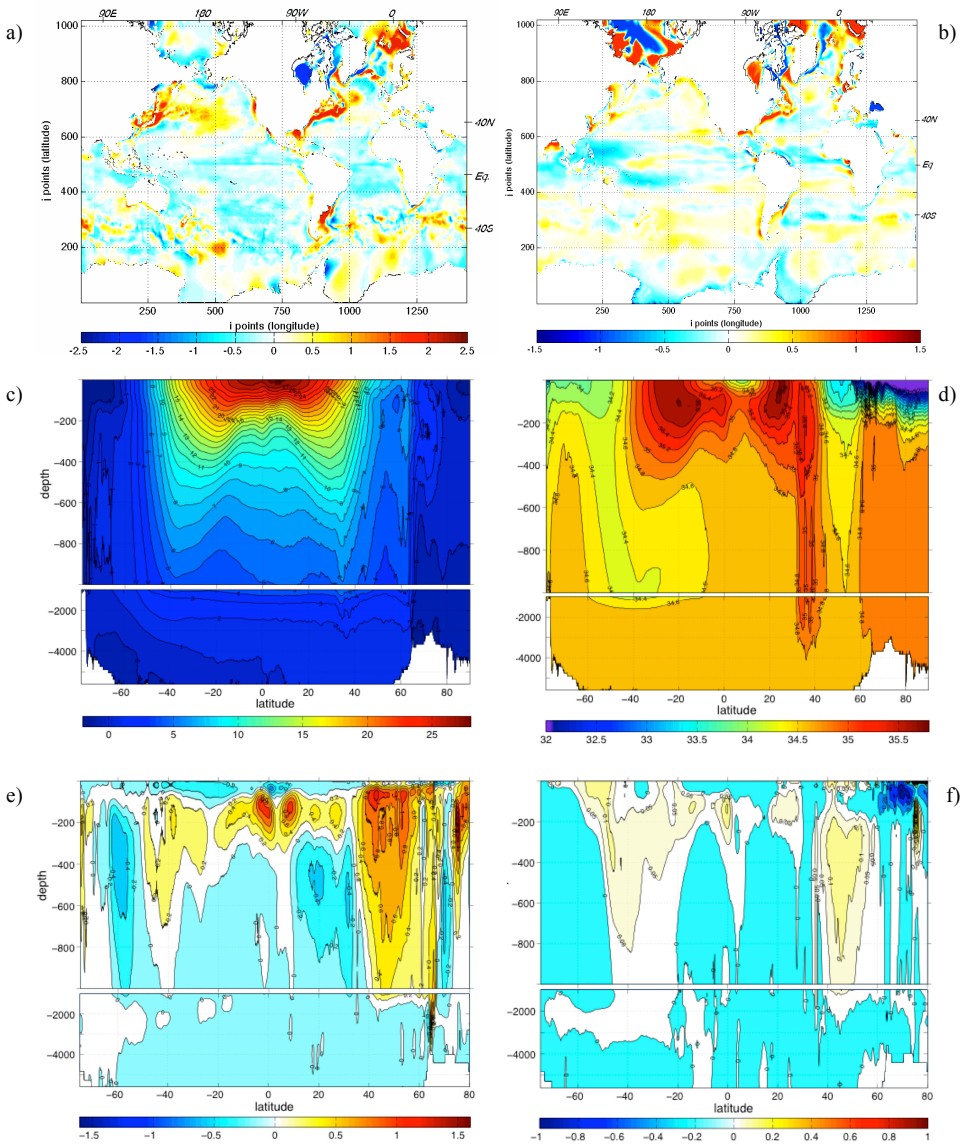

Fig. 2. (a, b) GLOB16 surface biases in years 2009-2013 for temperature and salinity. (c, d) Modelled zonal mean
temperature and salinity and (e, f) differences with EN3. Black and Caspian Sea are not considered in the zonal
mean. Temperature (salinity) is in the left (right) column. The contour interval is 0.2 ºC in (a,e), 1 ºC in (b), 0.05
psu in (b,f), 0.2 psu in (d). In (a,b) model output and observations are shown on the eddy-permitting ORCA grid.
Numbers of grid points are indicated on the axis, along with indications of latitudes and longitudes.

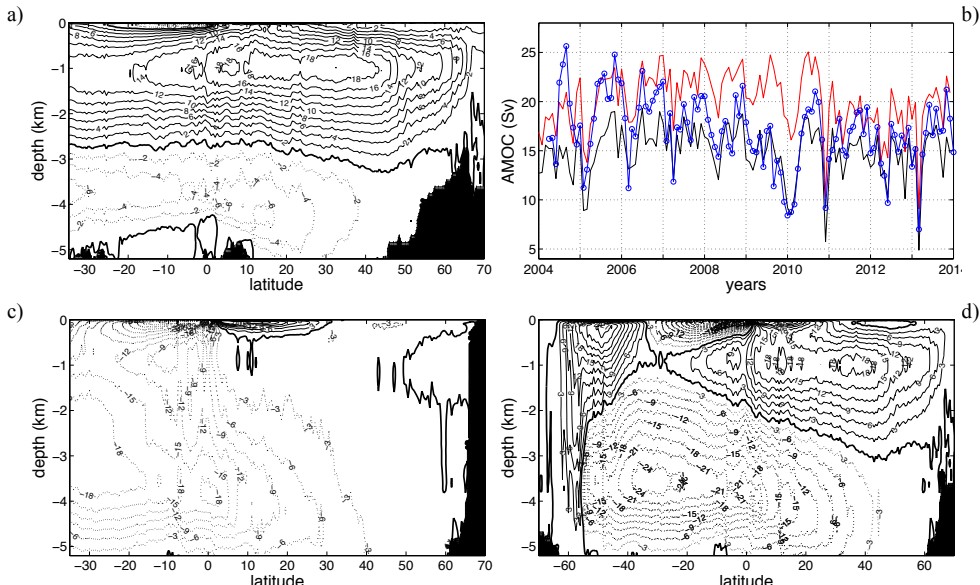

Fig. 3. Meridional overturning stream function (in Sv) averaged over the period 2009–2013 for (a) the Atlantic,
(c) the Indo-Pacific basins, and (d) the global ocean. The contour interval is 3 Sv. Thin solid lines represent
positive (clockwise) contours; thick solid lines represent zero contours. The stream functions were calculated with
0.5º latitudinal spacing to smooth out small-scale variations. (b) Time series of the AMOC at 26.5º N from
RAPID observational estimates (blue), GLOB16 (red) and GLOB4 (black) numerical simulations.



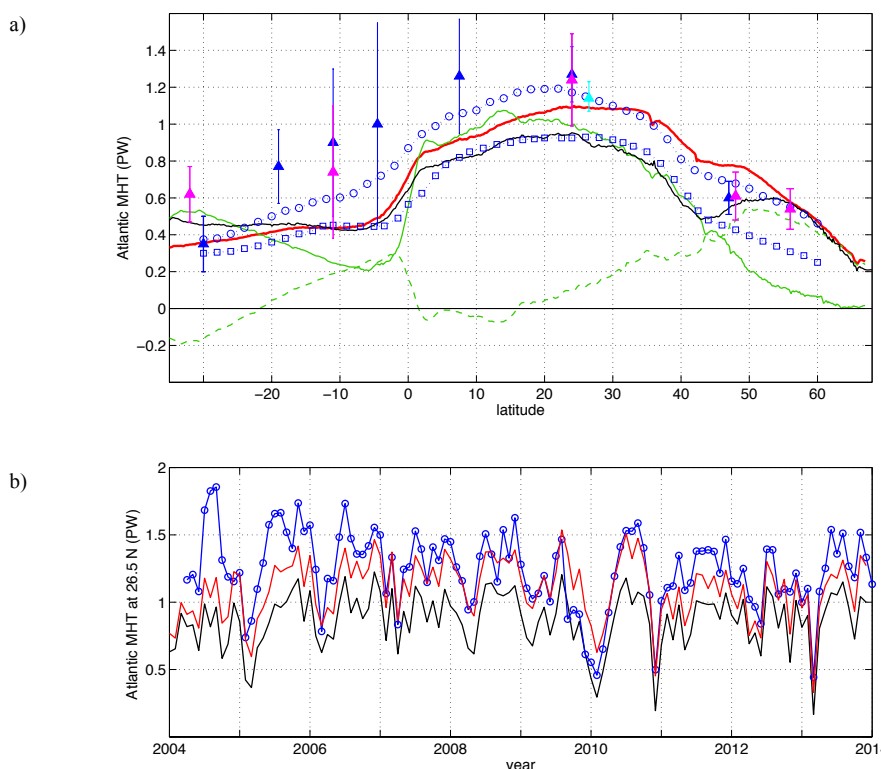

Fig. 4. (a) Time-mean Atlantic MHT (in PW) as a function of latitude. Red line is the total GLOB16 transport
with its overturning (green) and gyre (dashed green) and components. Black line represents the total GLOB4
transport. Blue circles (squares) represent implied time-mean transport calculated by Large and Yeager 2009
(Trenberth and Fasullo 2008). Triangles indicate direct estimates with their uncertainty ranges from the 2009-
2013 RAPID data (cyan), from Ganachaud and Wunsch 2003 (blue) and Lumpkin and Speer 2007 (magenta). (b)
Times series of the total Atlantic MHT across 26.5° N as estimated by RAPID (blue), from GLOB16 (red) and
GLOB4 (black).

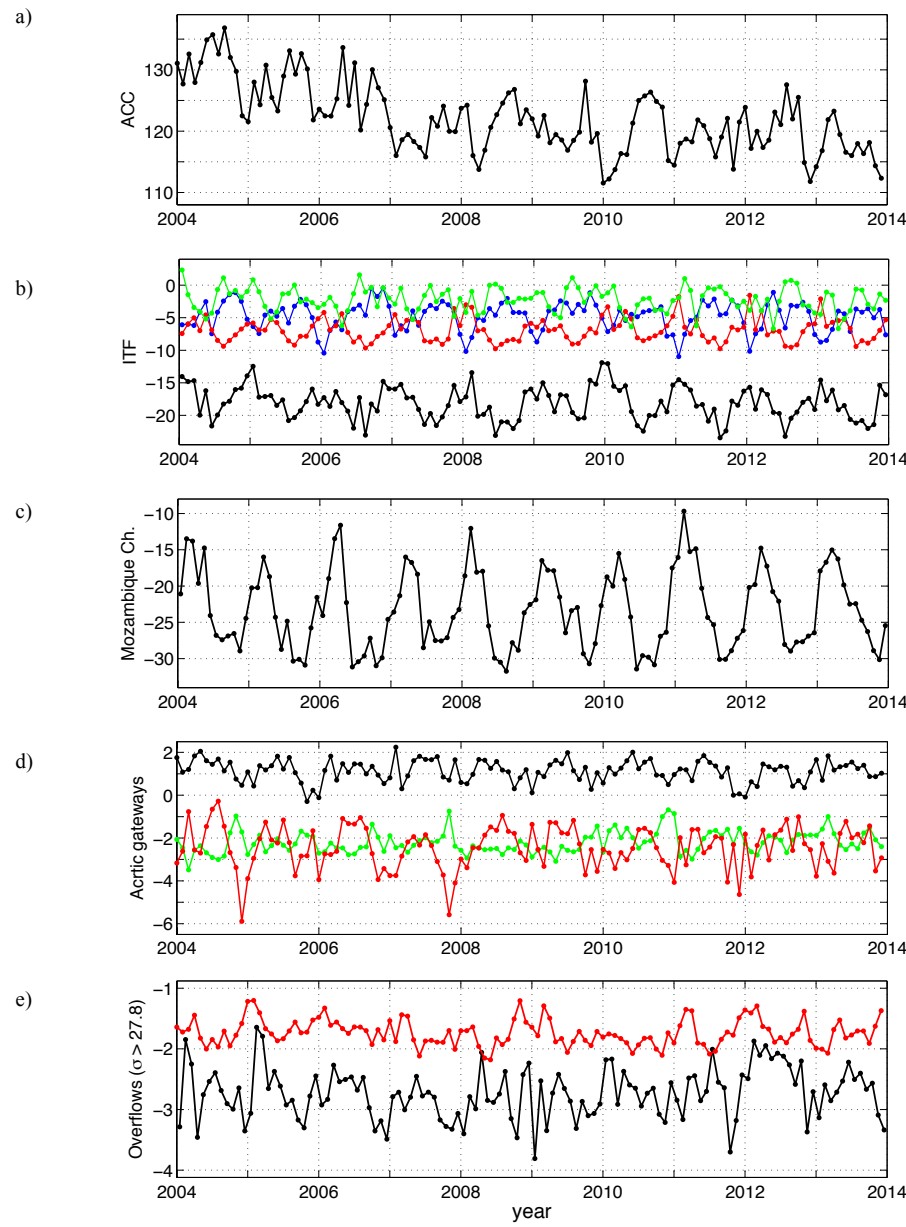

Fig. 5. Time series of the monthly averaged volume transport (in Sv) of the (a) ACC, (b) ITF (decomposed in Timor passage (red), Ombai strait (blue) and Lombok strait (green)), through (c) the Mozambique Channel, (d) Bering Strait (black), Fram Strait (red) and Davis Strait (green), and (e) for dense overflow through Denmark Strait (black), Faroe Bank Channel (red).





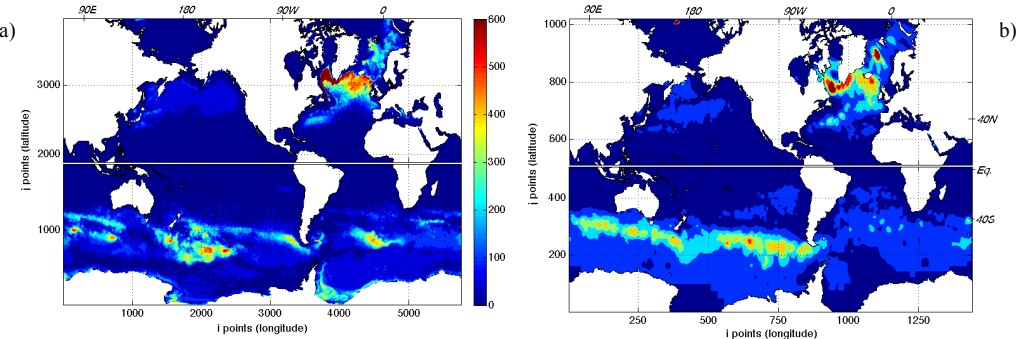

Fig. 6. (a) MLD (in m) averaged over March (in the Northern hemisphere) and September (in the Southern
hemisphere) 2009-2013 from (a) GLOB16, and (b) the de Boyer Montégut et al. (2004) climatology, based on
a 0.03 threshold on density profiles. Model output is shown on the GLOB16 grid; observations are interpolated on
the eddy-permitting ORCA grid. Numbers of grid points are indicated on the axis, along with indications of
latitudes and longitudes.

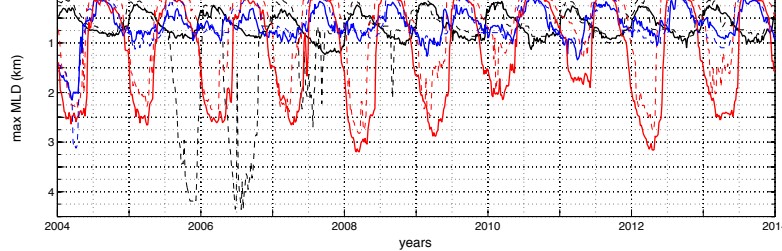

Fig. 7. Time series of modelled MLD maxima (in km) in the North Atlantic Ocean (red), the Nordic Seas (blue)
and the Southern Ocean (black) from GLOB16 (solid lines) and GLOB4 (dashed).



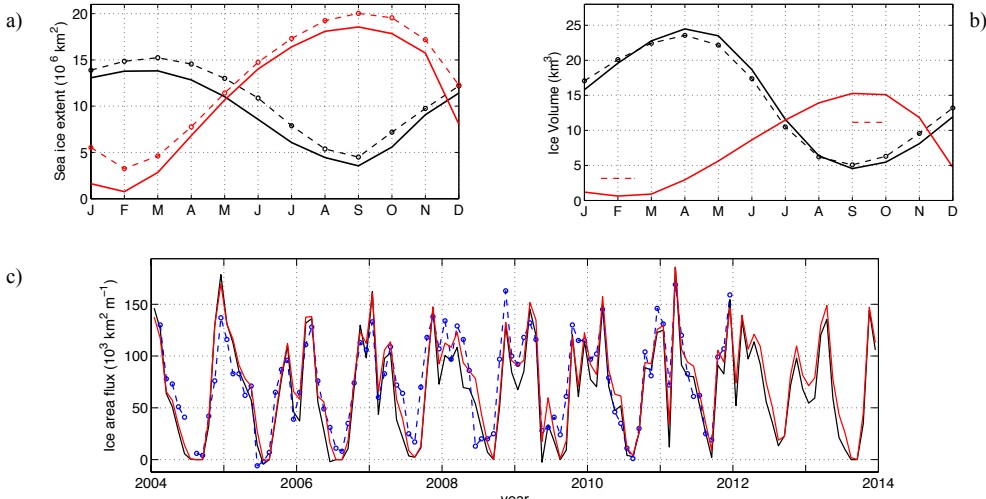

Fig. 8. (a) Mean GLOB16 seasonal cycles of sea ice extent ($10^6$ km$^2$) for the Arctic (black) and Antarctic (red) oceans compared to satellite observations (dashed line) provided by NSIDC. Sea ice extent is defined as the area enclosed in the 10% sea ice concentration contour. (b) Mean seasonal cycles of sea ice volume ($10^3$ km$^3$) for the Arctic Ocean (black) compared to PIOMAS reanalysis, and for the Antarctica (red) compared to minimum and maximum values from ICESat. (c) Sea ice area export ($10^3$ km$^2$ month$^{-1}$) across Fram Strait for GLOB16 (red), GLOB4 (black) and observations (blue).





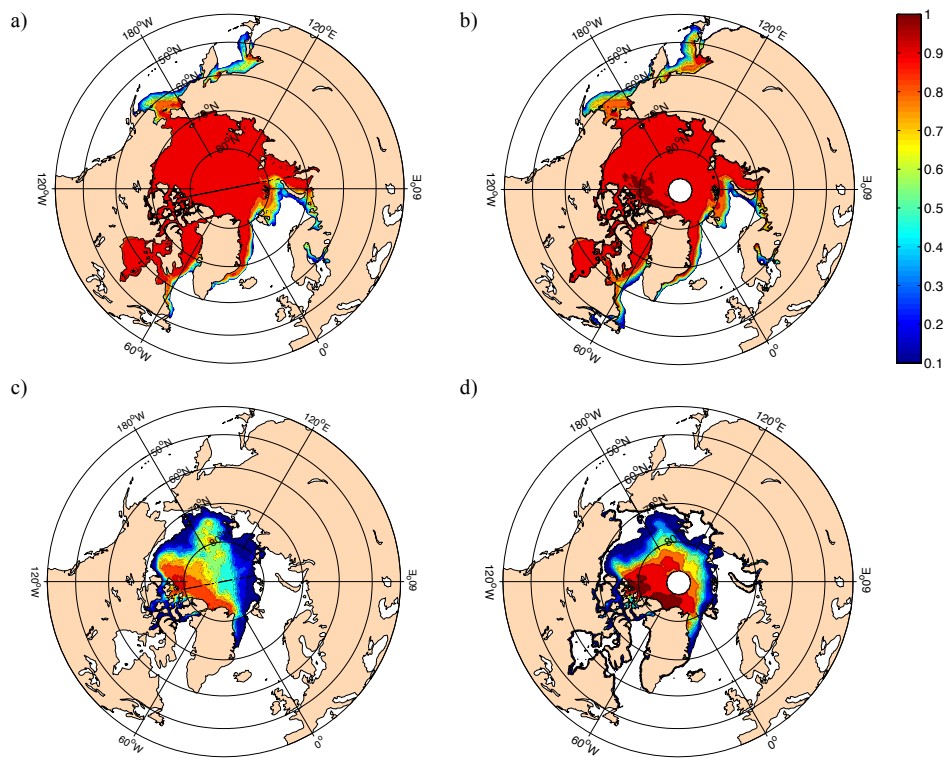

Fig. 9. Maximum (a, b) and minimum (c, d) Arctic sea ice concentration for the period 2009-2013 in GLOB16
(left) and observational data set (right).





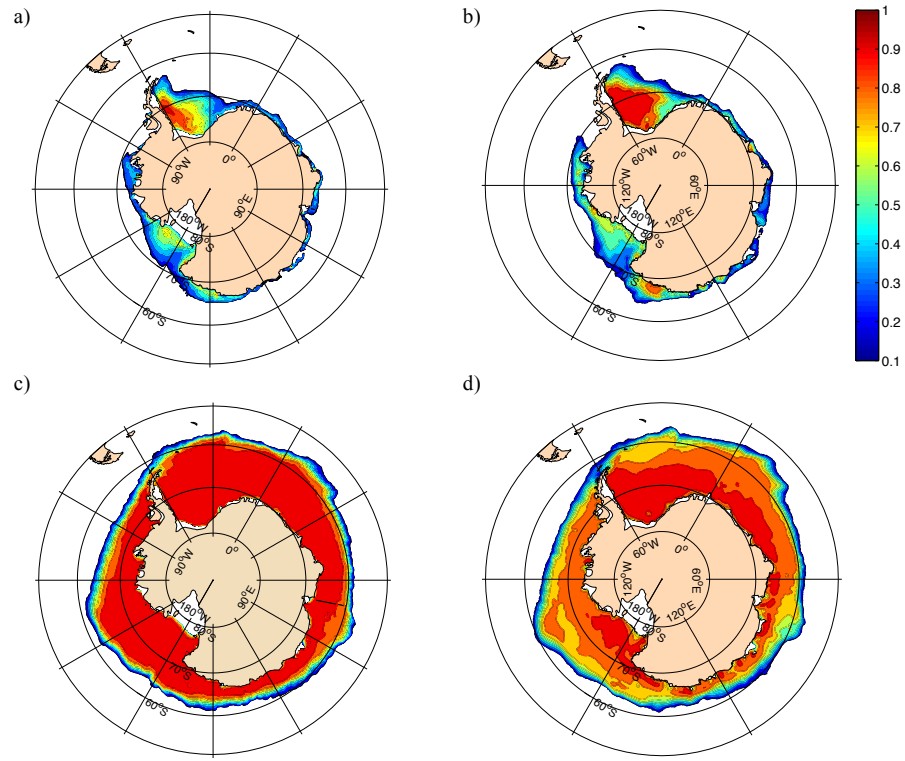

Fig. 10. Maximum (a, b) and minimum (c, d) Antarctic sea ice concentration for the period 2009-2013 in
GLOB16 (left) and observational data set (right).





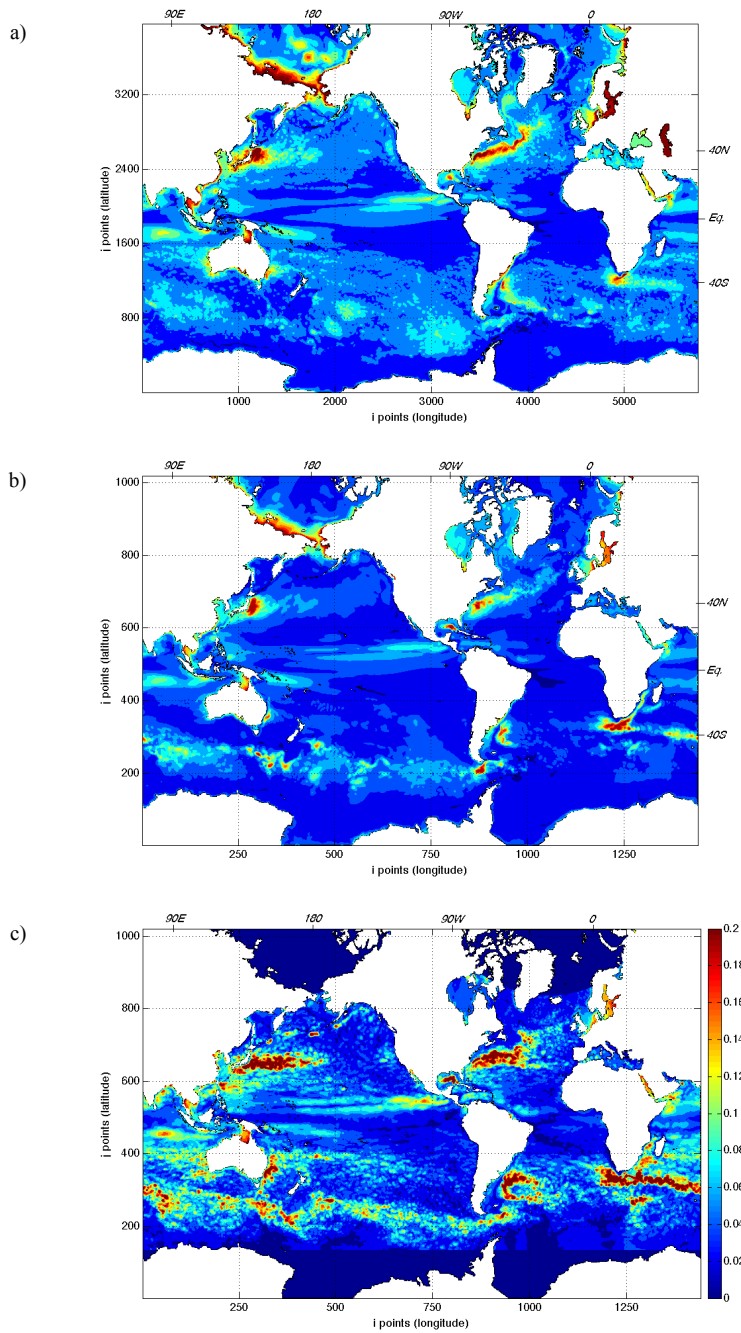

Fig. 11. Sea surface height variability (in m) from (a) the GLOB16 model, (b) the GLOB4 model and (c) AVISO. Modelled fields are shown on the own model grid; observations are interpolated on the eddy-permitting ORCA grid. Numbers of grid points are indicated on the axis, along with indications of latitudes and longitudes.





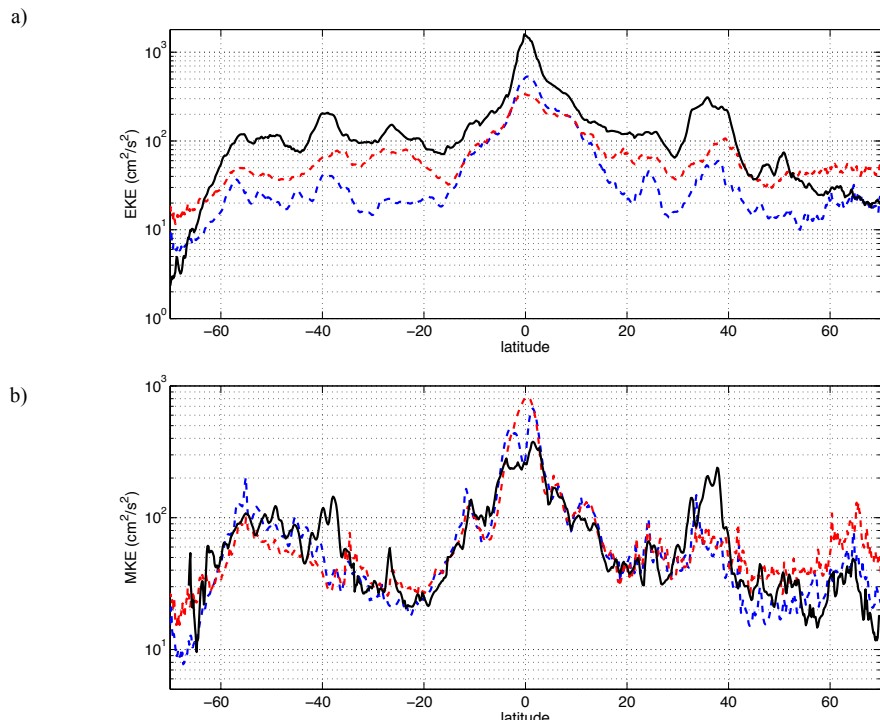

Fig. 12. (a) Latitudinal profiles of the global zonal-mean EKE (in cm$^2$ s$^{-2}$) of the surface flow for 2013 from
GLOB16 (red), GLOB4 (blue) and OSCAR (black). Scale is logarithmic. (b) As (a), but for the MKE of the
surface flow (in cm$^2$ s$^{-2}$).