# Peer review of "A 1/16° eddying simulation of the global NEMOv3.4 sea ice-ocean system"

_Geoscientific Model Development, 2015_

## Referee Comment (RC1) · Anonymous Referee #1 · 22 Mar 2016

General comments

This paper presents a summary of the basic performance for a ~ten year run of a new 1/16th global model, which is to be employed for operational ocean forecasts. Some performance comparisons are made with a similar 1/4 degree configuration of the model.

Overall I thought it was a well written paper which presented some interesting results. Most of the analyses are fairly standard, but to me this is as one would expect for a system definition paper.

My key comments are the paper could make better use of the $\frac{1}{4}$ degree parallel experiments they use and describe in places. For example:

• It is key to emphasize that resolution is not a panacea – there are many atmospheric forcing set errors, and errors due to limited physics, i.e. parameterisation, e.g. mixing in particular. There may also be error cancellation, e.g. a bias might get worse due to removing a cancelling error, e.g. with ocean forcing set errors, by increasing resolution.

• I would ideally prefer to see many more comparison sub-plots for existing figures to identify differences of the 1/12th compared to the $\frac{1}{4}$ twin run so we can see how many of the features you describe are common to both $\frac{1}{4}$ and 1/12 and how many are really down to

• The reader ideally needs to know exactly which parameters and config settings are different for the $\frac{1}{4}$ degree run compared to the 1/12th run. For example, is it the same version of NEMO, you mention it has level vertical levels so some differences could be due to vertical rather than spatial resolution. Also what coefficient for isopycnal mixing on tracers do you use at $\frac{1}{4}$ (as the $\frac{1}{4}$ simulation and biases are quite sensitive to this)?

Minor comments:

Line 42-43 I think it may depend on the definition but is there not a factor of pi between the Rossby radius (based on wave number) and eddy scale? This makes a difference as one only needs two grid cells per Rossby radius to get 6 cells per eddy. This would be worth clarifying?

Line 53 – should you mention 'often (wrongly) termed eddy resolving' in view of the fact you go on to state they are not eddy resolving at high latitudes as shown by Hallberg (2013)?

Line 71-74 – This is a long sentence. Also should you qualify this statement in view of your paragraph above, i.e. state we are now able to at least resolve eddies mostly equatorward of 50-60N/S BUT we don't resolve high latitude eddies or sub mesoscale or associated energy cascade anywhere. Furthermore, results are sensitive to grid

scale closure, particularly viscosity, as you state in your eddy kinetic energy section?

Line 88 – As stated above should it be described as a step forward, particularly for mid latitudes where this resolution resolves eddies but $\frac{1}{4}$ doesn't?

Line 123 – what about connection from Marmara sea to Aegean - Dardanelles Strait is very important for seasonal freshwater input to Northern Aegean.

Line 127 – Comment only – I believe some 1/12th NEMO configurations use partial slip in Labrador Sea to generate more eddies to help re-stratification after convection?

Line 141 – Comment only – important to note that uncorrected ERAI interim fields, e.g. radiation fields due to cloud errors, will have large errors which would be expected to impact on or even dominate near surface biases.

Line 158 – Why do you need to use SST restoration? This will mask model errors in near surface fields and there is already inherent relaxation back to air temperature in the forcing set?

Line 194 – I wonder why you didn't remove the seasonal cycle of MKE from EKE as otherwise the seasonal cycle of flows will be included in EKE estimates?

Line 215 – Can one really say much about SST biases from a ten year run with SST relaxation?

Line 215 – Should you show equivalent plots for figs 2 for $\frac{1}{4}$ degree or state that they are indistinguishable from the 1/12th if it is? The difference with the $\frac{1}{4}$ degree model is surely a key result?

Line 236 – Will the deep (>1000m) ocean really have equilibrated in a ten year run? I am guessing you are probably looking mainly as biases due to isopycnal heave which occurs reasonably fast?

Line 253/296 – More discussion or plots for GLOB4 on figs 3 and 4 would be useful?

[Figure]

Line 263 – Surely one can not say much about AABW in a ten year run?

Line 317 – Implied heat transports assume equilibrium? How large are you heat content tendencies with a short ten year run that may well still be drifting?

Line 430 – I did not think the location of mixed layer maxima agreed so well with the observations in Southern Ocean?

Line 430 – How does GLOB4 look in spatial plots? If it is very similar is it worth stating this?

Line 439 – I wonder if a different temperature based criterion might make the model mixed depths appear better? For example, if there is salinity compensation then density criterion can be rather sensitive to salinity errors.

Line 550 – I would emphasize the point about viscosity sensitivity more and include it in introduction. There is often an optimum viscosity level for EKE (and associated MKE) as too little enhances grid scale noise which damps eddies and too much obviously damps eddies.

Line 617 – Southern Ocean MLD maxima appear also not too good in Southern Ocean?

Line 633 – As it appears that you change both vertical and spatial resolution it is hard to conclusively attribute GLOB16 versus GLOB4 differences to spatial resolution. Could you state all configuration differences between GLOB4 and GLOB12 configurations as bullet points. In an ideal world one would mimimise the differences, e.g. use same number of vertical levels and vary only viscosity, isopyncal diffusion and perhaps slip between the two runs?

---

## Referee Comment (RC2) · Anonymous Referee #2 · 11 Apr 2016

This manuscript outlines the first results from a new global 1/16° implementation (GLOB16) of the NEMO-LIM ocean-sea ice model. The manuscript outlines some key metrics from the model and compares some metrics to a lower resolution implementations of the same model (GLOB4).

The model described in this manuscript is close to the leading edge of global ocean-sea ice models. It's important to document these models as they develop, and thus there are good reasons for GMD to want to publish this paper. However, there are a number of areas in which the manuscript could be improved.

My primary query is whether this manuscript is here simply to document the existence of a viable model (that is, the model works and is sufficient) or whether the aim is to make the case that the model is an improvement over previous, lower resolution versions. I strongly recommend following the latter path, but I found on reading the paper that the case for the GLOB16 model being an improvement on GLOB4 was somewhat tenuous. For many metrics the GLOB4 results were not shown, and in some areas GLOB4 looked slightly better! If one is to justify the move towards eddy-resolving models then a stronger case that the additional computational expense is worthwhilemust be built. (Alternatively, perhaps the conclusion may be that eddy-resolving is not worth the expense until models improve!) There are more details on these issues in the following list of suggested improvements that the authors may want to consider:

1. The use of acronyms (e.g. NEMO, CMCC) should be avoided in the abstract. In fact, CMCC is never defined in the text of the paper, and it seems unnecessary to list the affiliation of authors within the manuscript.

2. There is an ambiguous phrase on line 79: "... all (most of) the domain..." I suggest being more explicit.

3. In section 2.3, it's important to list more details about the magnitude of biharmonic viscosity, diffusivity, etc. If it's complicated, then a figure can be justified.

4. The SST restoring timescale seems very strong ... This value needs justification.

5. The first part of 2.7 should be shifted to 2.8. It also refers to an appendix which isn't present?

6. I don't understand the phrase bi-Laplacian (line 217). I'm used to either Laplacian or biharmonic.

7. On line 233 and beyond, replace TKE with simply KE (as many fields use TKE to represent Turbulent KE).

8. In 3.1, I'm not convinced that the mean surface biases mean anything in the presence of such strong restoring.

9. What is called AIW here is usually referred to as AAIW.

[Figure]

10. I found the results to be somewhat out of order. I suggest putting the global SSH variance maps second, right after the EKE results. In addition, I would put the global transport values before the AMOC results.

11. In 3.2, the depth-space overturning means very little in the Southern Ocean. The global MOC should be calculated in density space. The Deacon cell (line 350) is not a physically relevant cell and it would be better to estimate the size of the lower overturning cell in density space.

12. In Figures 2, 3, 5, 6, 8a,b, 9 and 10 there was no information on the GLOB4 results. However, occasionally, there were references in the text. As noted above, this manuscript will be much stronger if we know where and how improvements between GLOB4 and GLOB16 are manifested, so these results should be included wherever possible.

13. In Fig. 5 I would also like to see a line indicating estimates and errors of each quantity from observations. (Some are listed, some are not. In particular, the Mozambique Channel transport is stated as being "within the range of observed estimates" without a reference!) Also, the ACC transport, listed as the average over all years, is steady and very low for the last 6 years — it is this equilibrium value that should be listed, not the average over all years.

14. On line 444, it is ambiguous as to which "two transports are out of phase".

15. One open question which deserves more investigation is the lack of mesoscale variability in the Southern Ocean. There is a suggestion (in the Conclusion) that this is due to viscous parameterisations, but no quantitative information on what those parameterisations are. The Southern Ocean is one of the key locations where one might expect this resolution to make a dynamical difference, but the very low variability and ACC transport indicate that something is missing here. I suggest a deeper quantitative comparison with other high resolution models is in order.

---

## Author Response (AR1)

Dear Editor and Reviewer,

We would like to thank you for accurately reading and commenting the manuscript, and suggesting how to improve it. Answers to your comments are given in details hereafter. We hope that you will find them satisfactory. All authors agree with the modifications made to the manuscript.
Reviewer comments are in bold, and are followed by our response (in blue) that includes changes and/or additions to the text.

For the authors,
Dorotea Iovino

Answer to **Referee #1**

**General comments**
**This paper presents a summary of the basic performance for a _ten year run of a new 1/16th global model, which is to be employed for operational ocean forecasts.**
**Some performance comparisons are made with a similar 1/4 degree configuration of the model.**
**Overall I thought it was a well written paper which presented some interesting results.**
**Most of the analyses are fairly standard, but to me this is as one would expect for a system definition paper.**
**My key comments are the paper could make better use of the 1/4 degree parallel experiments they use and describe in places. For example:**
**\* It is key to emphasize that resolution is not a panacea – there are many atmospheric forcing set errors, and errors due to limited physics, i.e. parameterisation, e.g. mixing in particular. There may also be error cancellation, e.g. a bias might get worse due to removing a cancelling error, e.g. with ocean forcing set errors, by increasing resolution.**

We do agree that high horizontal resolution is not a panacea for all ocean modelling problems and it does not guarantee that all undesirable characteristics of our models are ameliorated. In fact, a number of prominent biases and model errors persist, or even worsen, despite increases in model resolution. The increase in horizontal and vertical resolutions needs to be accompanied by improved modelling of relevant physical processes at the appropriate scale. Model parameterizations that have been developed for coarse resolutions mat not be ideal for considerably finer spatial-scales and may need to be revised, requiring model developments. However, the finer resolution remains one possible way in which model capabilities can be enhanced, thanks to the explicit solution of eddies. In this paper, we want to document if and how an "eddy-resolving" ocean model resolution impacts the simulation of large-scale ocean variability with respect to an eddy-permitting configuration. We do believe that our GLOB16 results, in accordance to the current literature, provide compelling evidence that an eddying ocean component can significantly impact the simulation of the large-scale ocean dynamics. We modified the Introduction to account for this comment: "Although the increase of resolution does not necessarily lead *per se* to an improved representation of the ocean general circulation, the aim of this work is to evaluate the effect of the explicit solution of eddy dynamics at low- and mid- latitudes on the large-scale dynamics of a high-resolution global ocean model, compared to a coarser resolution configuration".

**\* I would ideally prefer to see many more comparison sub-plots for existing figures to identify differences of the 1/12th compared to the 1/4 twin run so we can see how many of the features you describe are common to both 1/4 and 1/12 and how many are really down to**
We agree with the referee that a more detailed comparison between GLOB16 and GLOB4 can definitely improve the manuscript. To present a more complete comparison with GLOB4, we modified subplots and/or add text in any relevant part of section 3, trying not to alter too much the manuscript structure or increase its length.

**\* The reader ideally needs to know exactly which parameters and config settings are different for the 1/4 degree run compared to the 1/12th run. For example, is it the same version of NEMO, you mention it has level vertical levels so some differences could be due to vertical rather than spatial resolution. Also what coefficient for isopycnal mixing on tracers do you use at 1/4 (as the 1/4 simulation and biases are quite sensitive to this)?**
We added the key differences in parameters and configuration settings at the end of section 2.

**Minor comments:**
**Line 42-43 – I think it may depend on the definition but is there not a factor of pi between the Rossby radius (based on wave number) and eddy scale? This makes a difference as one only needs two grid cells per Rossby radius to get 6 cells per eddy. This would be worth clarifying?**
The direct relation between the Rossby radius, R, and the baroclinic Rossby wavelength, λ, is R = λ/2π. The grid spacing Δ has to satisfy Δ < R, and then 2πΔ < λ, following the traditional criteria for resolving a wave on a discrete grid. Nevertheless, Hallberg (2013) considered a 2Δ < R criteria.
In order to clarify this, we modified the text in section 1 and added the following lines:
"Hallberg (2013) showed the model horizontal resolution required to resolve the first baroclinic deformation radius with two grid points, based on a Mercator grid. From his analysis, 1/4° Mercator spacing is insufficient to resolve mesoscale eddies that have a typical scale of 50 km at mid-latitudes."

**Line 53 – should you mention 'often (wrongly) termed eddy resolving' in view of the fact you go on to state they are not eddy resolving at high latitudes as shown by Hallberg (2013)?**
We apologise for pointing out that we did find this specific text neither in line 53 nor elsewhere in the manuscript. As reported by emails, we did not detect a direct correspondence between some line numbers indicated by the Referee's comments and the line numbers reported in the pdf file (available online http://www.geosci-model-dev-discuss.net/gmd-2015-268/gmd-2015-268.pdf). However, following this comment, we highlighted in the introduction that GLOB16 and configurations with comparable horizontal resolution do not resolve mesoscale globally, and are only eddy-permitting north of a certain latitude and over shelf regions. The test now reads "Resolving mesoscale eddy variability remains anyway elusive at higher latitudes. For example, in the Arctic Ocean where the first Rossby radius decreases down to few kilometres on the continental shelf or in weakly stratified regions, typical eddy-resolving resolution does only permit eddies at best (Nurser and Bacon 2014)."

**Line 71-74 – This is a long sentence. Also should you qualify this statement in view of your paragraph above, i.e. state we are now able to at least resolve eddies mostly equatorward of 50-60N/S BUT we don't resolve high latitude eddies or sub mesoscale or associated energy cascade anywhere. Furthermore, results are sensitive to grid scale closure, particularly viscosity, as you state in your eddy kinetic energy section?**
In accordance with the Referee's suggestion, text has been added to specify where the 1/16° resolution is not eddy-resolving: "In this context, we developed a global eddying configuration, where eddying means that the numerical simulation is eddy-resolving in most deep ocean regions equatorward of 60°, while it is mostly only eddy-permitting at higher latitude."

**Line 88 – As stated above should it be described as a step forward, particularly for mid latitudes where this resolution resolves eddies but 1/4 doesn't?**
We do confirm that, as stated in the abstract, GLOB16 configuration represents a step forward in the CMCC global ocean modelling to resolve eddies in the ocean at mid-latitudes. At line 88, we want to emphasise also that this configuration is, for our modelling group, an accomplishment that open the way toward a new, operational short-term ocean forecast system. We did slightly modify the sentence in "…is a foothold that opens the way for the development of a new, operational short-term ocean forecast system…"

**Line 123 – what about connection from Marmara Sea to Aegean - Dardanelles Strait is very important for seasonal freshwater input to Northern Aegean.**
We did not mention the Dardanelles Strait into the bathymetry description (section 2.2) because no specific modification was applied there. After interpolating GEBCO dataset on the GLOB16 grid, the Dardanelles strait resulted open and 3-grid-point wide in its narrowest area. We reasoned that additional hand editing was unnecessary in that location. On the other hand, the Bosphorus strait (with a maximum width of ~3.5 km) was close and we had to modify the bathymetry to connect the Marmara Sea and the Black Sea.

**Line 127 – Comment only – I believe some 1/12th NEMO configurations use partial slip in Labrador Sea to generate more eddies to help re-stratification after convection?**

This comment refers to ORCA12 configuration. We are aware of a set of sensitivity tests performed by the DRAKKAR group to study the impact of lateral boundary condition on the mixed layer depth (MLD) and eddy kinetic energy (EKE) in the Labrador Sea, the Mediterranean Sea and other location where the boundary current dynamics is relevant. Free slip, partial slip and "variable (local partial)" slip were considered. In the Labrador Sea, for example, their results showed that local partial slip helps to increase EKE with its maximum well located between 60-62°N, and also reduce MLD. They suggested that the best choice is a combination of free slip applied everywhere except few limited patches with no-slip. Those locations have to be identified in order to build a sort of mask. We do agree that this approach might help to improve the GLOB16 results in some areas. A set of improvements is already planned for this configuration, some of them already employed. The lateral boundary conditions are for sure on the list.

**Line 141 – Comment only – important to note that uncorrected ERAI interim fields, e.g. radiation fields due to cloud errors, will have large errors which would be expected to impact on or even dominate near surface biases.**

We do agree with the referee that errors in the atmospheric forcing may largely affect surface biases. A set of corrections can be applied to ECMWF ERAinterim variables to reduce global and regional biases. For example, in developing the DRAKKAR Forcing Set, Brodeau et al. (2014) corrected ERAinterim winds (weaker than QuikSCAT in the inter-tropical band between 40°S and 40°N), modified air temperature and humidity in the polar regions, reduced the shortwave radiation and increased the longwave radiation, and corrected the precipitation field in the western tropical Atlantic and Pacific oceans. How those corrections can improve the forced simulation has been addressed. Nevertheless, we decided not to follow this approach, but rather using some surface restoration (see reply to next point). Our decision may be justified by a number of arguments. Our ocean-modelling group generally uses the uncorrected-ERAinterim atmospheric reanalysis, conducting an analysis of the effect of the quality of the atmospheric forcing on the high-resolution system was beyond the scope of this study, and this GLOB16 simulation has been used as bedrock for the CMCC global ocean forecast system that uses global ECMWF operational system and forecast as it is released. Note that applying corrections to operational analyses/forecasts is rather dangerous, due to possible unexpected changes in quality of the real-time atmospheric forcing.

**Line 158 – Why do you need to use SST restoration? This will mask model errors in near surface fields and there is already inherent relaxation back to air temperature in the forcing set?**

The SST relaxation is implemented for a twofold reason: to limit the propagation of the atmospheric forcing biases onto the upper ocean and to compare the results with the twin experiment GLOB4 at lower resolution. The value of the relaxation time-scale was set equal to that used in the 1/4° reanalysis and simulation system (Storto et al., 2016; Haid et al. submitted to The Cryosphere).

However, we acknowledge that, due to the SST relaxation, the verification of the sea surface temperature may be misleading and driven by the relaxation dataset and time-scale. Thus, in the revised version of the manuscript, we focused our analysis (in section 3.1) on the subsurface biases.

*Storto, A., S. Masina, and A. Navarra (2016), Evaluation of the CMCC eddy-permitting global ocean physical reanalysis system (C-GLORS, 1982–2012) and its assimilation components. Q.J.R. Meteorol. Soc., 142: 738–758. doi: 10.1002/qj.2673*

**Line 194 – I wonder why you didn't remove the seasonal cycle of MKE from EKE as otherwise the seasonal cycle of flows will be included in EKE estimates?**

Line 194 falls within section 2.7 on the "Output and analysis strategy". We assumed the referee's comment is about the calculation of EKE in section 3. As suggested, the seasonal cycle was estimated and then removed from the time series (Plot 1).

[Figure]

Plot 1. (Fig. 1b in the manuscript) Time variations of volume-averaged EKE (in $cm^2 s^{-2}$), where the black line represents the global basin-mean value and the red (blue) the contribution of the Southern (Northern) Hemisphere in GLOB16. Thin-dashed line represents the basin-mean EKE in GLOB4.

**Line 215 – Can one really say much about SST biases from a ten year run with SST relaxation?**

We do agree with the referee. Due to the length of the simulation and the temperature restoring that we applied at the sea surface, the analysis of the SST biases does not add much to our study. We removed the plots showing the surface biases (Fig 2a, b) and accordingly correct the text at the beginning of section 3.1 as follow "The mean fields of modelled potential temperature and salinity are here validated against the mean of EN3 climatology (the UK Met Office Hadley Centre observational dataset, Ingleby and Huddleston 2007), both averaged over the same period 2009-2013. As expected, due to the temperature and salinity restoring applied at the ocean surface, the global mean SST and SSS biases are small (-0.06 for SST and -0.04 for SSS). There are weak cold biases in the tropics, extending over much of the subtropical band, with the largest SST biases (~1 °C warmer) collocated with positive SSS error (0.5–1.5 psu) over the western boundary currents in the Atlantic and North Pacific oceans (not shown). The overall pattern of surface biases is similar between the two models."

**Line 215 – Should you show equivalent plots for figs 2 for 1/4 degree or state that they are indistinguishable from the 1/12th if it is? The difference with the 1/4 degree model is surely a key result?**

We followed this comment and we added two subplots (e, f) in Figure 2 showing the zonal mean temperature and salinity differences between GLOB4 and observations. Section 3.1 includes now the following text "Although the overall biases are similar between the two model configurations in many latitude bands, there are some relevant differences (Fig. 2e, f). For instance, the Southern Ocean is generally warmer in GLOB4, with a larger positive salinity bias at ~400 m depth around 50°S. Both models are warm and saline in the above depth range in the northern mid and high latitudes, but the biases differ in magnitude and locations, highlighting the difference in path of the western boundary currents. Both models are warmer than observations in the Artic Ocean: the largest warming is confined in the upper 200 m depth in GLOB16, while the maximum, with a similar value, is located between 300 and 500 m depth in GLOB4."

**Line 236 – Will the deep (>1000m) ocean really have equilibrated in a ten year run? I am guessing you are probably looking mainly as biases due to isopycnal heave which occurs reasonably fast?**

We do agree with the Referee that the integration time required to "spin up" an ocean model from an initial state of rest to a near equilibrium state is much longer for the assessment of the deep ocean. Unfortunately, obtaining equilibrium at eddying resolution is not practical. In 10-year integration, GLOB16 is still in its adjustment phase, especially in the deep ocean. This short simulation is not appropriate for studying the long-term evolution of deep-water masses, as we assert in section 2.6, "The model run for 11 years through the end of 2013, which appears to be a sufficient amount of time for the near-surface velocity field to adjust to the initial density field and for mesoscale processes in the upper ocean to have reached a quasi-equilibrium, while the deep ocean takes much longer to reach steady state." In the paper, we are limited to describe the state of the deep ocean hydrography and overturning circulation after 10 years of integration.

**Line 253/296 – More discussion or plots for GLOB4 on figs 3 and 4 would be useful?**
Figure 3 includes the time series of AMOC at 26.5°N as reproduced by GLOB4, together with GLOB16 output and RAPID estimates. These results are discussed in section 3.2. We decided not to add plots of the MOC stream functions in the Atlantic, Indo-Pacific and Southern oceans for GLOB4, but we added descriptions of the differences in the overturning circulation between the two configurations for each basin. Both subplots in Figure 4 included GLOB4 results (black lines) as the time-mean Atlantic MHT and its 10-y time series at 26.5°N, also in the previous version of the manuscript. A more detailed description has been added.

**Line 263 – Surely one can not say much about AABW in a ten year run?**
We do agree that in GLOB16 we cannot evaluate the performances of the model to reproduce the deep ocean circulation in a 10-year run. We limit our purpose to providing estimate of the mean model transport in the bottom layer in such a short simulation.

**Line 317 – Implied heat transports assume equilibrium? How large are you heat content tendencies with a short ten year run that may well still be drifting?**
One caveat of this study is that the simulations lasted only 10 years.
Comparing actual ocean heat transports with those implied by surface fluxes gives an indication of the volume-averaged drift in temperature. The time evolution of the volume-averaged ocean temperature is shown in Fig 1c to demonstrate the extent to which a quasi-steady state has been reached at the end of the 10-year integration. The potential temperature is seen to decrease by ~0.005°C over 10 years. This drift (related to discrepancies between the actual heat transports by the ocean and the heat transport implied by the surface fluxes) is small enough to assume a close agreement between implied and actual meridional heat transports. That suggests that the GLOB16 ocean is close to quasi-equilibrium.
Then, the time series of the MHT in the Atlantic Ocean at 26°N (at other selected latitudes in the Atlantic Ocean as well – not shown in the paper) does not present any particular trend over the 10-year time window.

**Line 430 – I did not think the location of mixed layer maxima agreed so well with the observations in Southern Ocean?**
Line 430 is in section 3.3 and concerns the transport in the Mozambique Channel. We think that this comment refers to section 3.4 where locations of mixed layer maxima were mentioned in lines 476-477 (following line numbers in previous section). We deleted this sentence and we kept line 489-490 in which we state that maxima in the Southern Ocean Pacific sector are "not exactly collocated with the observed ones".

**Line 430 – How does GLOB4 look in spatial plots? If it is very similar is it worth stating this?**
Figure 6 includes now a subplot showing GLOB4 mixed later depth averaged in the Northern (Southern) in March (September) 2009-2013.

**Line 439 – I wonder if a different temperature based criterion might make the model mixed depths appear better? For example, if there is salinity compensation then density criterion can be rather sensitive to salinity errors.**

We applied different criteria to our model output to compute the mixed layer depth. The differences are mainly local and we did not find any general improvement or worsening using different MLD calculations in both hemispheres.

**Line 550 – I would emphasize the point about viscosity sensitivity more and include it in introduction. There is often an optimum viscosity level for EKE (and associated MKE) as too little enhances grid scale noise which damps eddies and too much obviously damps eddies.**

A clear weakness of this first GLOB16 experiment is its ability in reaching the observed magnitude of EKE, especially in the Southern Ocean. This behaviour is most likely related to the coefficient chosen for lateral momentum diffusion. This GLOB16 simulation suggests that more efforts shall be dedicated to sensitivity experiments for detecting the optimal configuration of horizontal and vertical dynamics. Unfortunately, re-running 10-year experiments is not faceable at the moment. To improve this aspect of GLOB16, we are currently performing short (2year) test experiments.

**Line 617 – Southern Ocean MLD maxima appear also not too good in Southern Ocean?**

Already answered (please see above the comment to line 430)

**Line 633 – As it appears that you change both vertical and spatial resolution it is hard to conclusively attribute GLOB16 versus GLOB4 differences to spatial resolution. Could you state all configuration differences between GLOB4 and GLOB12 configurations as bullet points. In an ideal world one would mimimise the differences, e.g. use same number of vertical levels and vary only viscosity, isopyncal diffusion and perhaps slip between the two runs?**

Key differences in parameter settings between the two configurations are now listed in the end of section 2.

Vertical resolution may be adequate for the horizontal scales one is physically concerned with. Lindzen and Fox-Rabinovitz (1989) showed the substantial benefits in refining both horizontal and vertical resolution give some support to scaling arguments deduced from quasi-geostrophic theory implying that horizontal and vertical resolution ought to be chosen consistently. They developed simple physical criteria, based on Rossby radius and gravity wave dispersion, for the vertical resolution consistent with horizontal resolution.

Increasing the vertical resolution in the ocean model can provide a better representation of the upper ocean physics, improve the properties of dense water formation as well as the dynamics of overflow between ocean basins, and more accurately represent the bottom boundary layer physics. The DRAKKAR group showed, for example, improvements in ORCA12 representation of overflows when the vertical levels are changed from 75 to 300.

Both horizontal and vertical resolutions are needed to more accurately depict the 3D structure of the ocean. A multi-step approach to isolate the effects of any specific modification from GLOB4 to GLOB16 was, unfortunately, not viable. Therefore, GLOB16 includes both vertical and horizontal refined grids, and there are no twin runs in which the two configurations do share the same vertical discretization. We attempted to achieve a consistency between horizontal and vertical resolution in our GLOB16 configuration. Applying what we considered to be the most adequate choice (also considering the real limitation due to computational cost), we moved from 75 vertical levels in GLOB4 to 98 in GLOB16, with level spacing from 1 m near the surface to 160 in the deep ocean.

*Lindzen, R. S., and M. S. Fox-Rabinovitz, 1989: Consistent vertical and horizontal resolution. Mon. Wea. Rev., 117, 2575– 2583.*

**We have made the following additional changes to the paper:**
In the list of References, we updated Stepanov et al. 2016, Storto et al. 2016.

We corrected the Mozambique Channel transport in Table 2 and accordingly added the following reference
Ridderinkhof, H., P. M. van der Werf, J. E. Ullgren, H. M. van Aken, P. J. van Leeuwen, and W. P. M. de Ruijter: Seasonal and interannual variability in the Mozambique Channel from moored current observations, J. Geophys. Res., 115, C06010, 2010.

Dear Editor and Reviewer,

We would like to thank you for accurately reading and commenting the manuscript, and suggesting how to improve it. Answers to your comments are given in details hereafter. We hope that you will find them satisfactory. All authors agree with the modifications made to the manuscript.
Reviewer comments are in bold, and are followed by our response (in blue) that includes changes and/or additions to the text.

For the authors,
Dorotea Iovino

Answer to **Referee #2**

**This manuscript outlines the first results from a new global 1/16° implementation (GLOB16) of the NEMO-LIM ocean-sea ice model. The manuscript outlines some key metrics from the model and compares some metrics to a lower resolution implementations of the same model (GLOB4).**
**The model described in this manuscript is close to the leading edge of global ocean-sea ice models. It's important to document these models as they develop, and thus there are good reasons for GMD to want to publish this paper. However, there are a number of areas in which the manuscript could be improved.**
**My primary query is whether this manuscript is here simply to document the existence of a viable model (that is, the model works and is sufficient) or whether the aim is to make the case that the model is an improvement over previous, lower resolution versions. I strongly recommend following the latter path, but I found on reading the paper that the case for the GLOB16 model being an improvement on GLOB4 was somewhat tenuous. For many metrics the GLOB4 results were not shown, and in some areas GLOB4 looked slightly better! If one is to justify the move towards eddy-resolving models then a stronger case that the additional computational expense is worthwhile must be built. (Alternatively, perhaps the conclusion may be that eddy-resolving is not worth the expense until models improve!)**

The main objective of this study is to present a new global eddying configuration, to evaluate the first 10-year simulation and, not least, to show the overall improvements of the model solution due to the increasing resolution. For this aim, together with the validation of GLOB16 against observations, we presented a comparison between GLOB16 and the eddy-permitting configuration, GLOB4. As point out by the Referee, this study is certainly enriched by a more detailed analysis of the differences between the two models. We added text and modified plots in any subsection on "model validation" to emphasize the key role played by the ocean resolution.
GLOB16 simulation is by no means perfect, with notable discrepancies with observations in some areas, but we believe that our analysis demonstrates that most aspects of the GLOB16 circulation are more realistic with respect to GLOB4, and the eddying models is able to better represent the upper ocean dynamics and ocean variability at mid- and low latitudes. In particular, our analysis shows that eddying resolution improves the temperature and salinity biases at upper and intermediate depths, the extension and separation of western boundary currents, the strength of the overturning circulation as well as the poleward heat transport in the Atlantic Ocean, the volume transports through most of the considered critical passages, and the narrow boundary currents and flow over narrow sills. Tables 1 and 2 summarize the changes in volume transports.
The biggest caveat of this GLOB16 simulation is the representation of the mesoscale variability. The model has too weak eddy energy compared to observations, and the expected increase in eddy kinetic energy (EKE) due to resolution is not evident. In order to identify the source of this problem, we are analysing this aspect in detail performing a set of short (2-year) sensitivity experiments.

**There are more details on these issues in the following list of suggested improvements that the authors may want to consider:**

**1. The use of acronyms (e.g. NEMO, CMCC) should be avoided in the abstract. In fact, CMCC is never defined in the text of the paper, and it seems unnecessary to list the affiliation of authors within the manuscript.**
We deleted the CMCC acronyms in the abstract, substituting it with "Euro-Mediterranean Center on Climate Change". We do think that the acronym NEMO is well known in the ocean modelling community, and as suggested by the editor, we included it in the title and abstract. We deleted the version of the code that now appears only within the model description in section 2.

**2. There is an ambiguous phrase on line 79: ". . . all (most of) the domain. . ." I suggest being more explicit.**
We changed the sentence in "… the numerical simulation is eddy-resolving in most deep ocean regions equatorward of 60°, while it is mostly eddy-permitting at higher latitude and over shelf regions."

**3. In section 2.3, it's important to list more details about the magnitude of biharmonic viscosity, diffusivity, etc. If it's complicated, then a figure can be justified.**
We emphasised the differences in parameter settings between the two configurations at the end of section 2.

**4. The SST restoring timescale seems very strong . . . This value needs justification.**
The SST relaxation is implemented to limit the propagation of the atmospheric forcing biases into the upper ocean, and thus, with this constrain, reproduce a fairly realistic variability of the upper ocean heat content. The air-sea heat fluxes correction induced by the SST relaxation is shown to drive the strength of deep convection in crucial areas such as Labrador Sea and Nordic Seas, thus also positively impacting the strength of the meridional circulation (Storto et al. 2016).
The value of the relaxation time-scale was set equal to that used in the 1/4° reanalysis and simulation system (Storto et al. 2016; Haid et al. submitted to The Cryosphere) and is chosen as trade-off between the daily frequency of the SST analyses, without strongly constraining the air-sea heat fluxes themselves.
However, we acknowledge that due to the SST relaxation the verification of the sea surface temperature may be misleading and driven by the relaxation dataset and time-scale. Thus, in the revised version of the manuscript we focused our analysis (in section 3.1) on the subsurface biases.

*Storto, A., S. Masina, and A. Navarra (2016), Evaluation of the CMCC eddy-permitting global ocean physical reanalysis system (C-GLORS, 1982–2012) and its assimilation components. Q.J.R. Meteorol. Soc., 142: 738–758. doi: 10.1002/qj.2673*

*Haid, V., D. Iovino, and S. Masina. Impacts of Antarctic runoff changes on the Southern Ocean sea ice in an eddy-permitting sea ice-ocean model. Submitted to The Cryosphere*

**5. The first part of 2.7 should be shifted to 2.8. It also refers to an appendix which isn't present?**
Accordingly to reviewer's suggestions, we removed the text in the subsection 2.7 and moved into subsection 2.8, which has been partially rewritten to highlight the differences between the two configurations and avoid repetition in the set-up description. By mistake, we were still referring to an appendix, initially used to describe the eddy-permitting configuration, and then removed, as suggested by the editor. The descriptions of both models are included in section 2.

**6. I don't understand the phrase bi-Laplacian (line 217). I'm used to either Laplacian or biharmonic.**
We replaced the word bi-Laplacian by biharmonic.

**7. On line 233 and beyond, replace TKE with simply KE (as many fields use TKE to represent Turbulent KE).**
Thank you for this comment. We wrongly used the acronym TKE to identify both the Turbulent Kinetic Energy the Total Kinetic Energy. In the revisited version, TKE stays for Turbulent Kinetic Energy in section 2, while "total KE" refers to the total kinetic energy in section 3.

**8. In 3.1, I'm not convinced that the mean surface biases mean anything in the presence of such strong restoring.**

We do agree with the Referee and we removed the plots showing SST and SSS biases (Fig. 2a and b in previous version). Accordingly, we modified text in section 3.1 as follow "As expected, due to the temperature and salinity restoring applied at surface, the global mean SST and SSS biases are small (-0.06 for SST and -0.04 for SSS). There are weak cold biases in the tropics, extending over much of the subtropical band, with the largest SST biases (over 1 °C) collocated with positive SSS error (0.5–1.5 psu) over the western boundary currents in the Atlantic and North Pacific oceans (not shown)."

**9. What is called AIW here is usually referred to as AAIW.**

Sorry for this oversight. We corrected the Antarctic Intermediate water acronym in AAIW.

**10. I found the results to be somewhat out of order. I suggest putting the global SSH variance maps second, right after the EKE results. In addition, I would put the global transport values before the AMOC results.**

We thank the referee for the comment, but we do prefer to keep the actual structure. The global SSH variance is a weakness (probably the largest) of this first GLOB16 experiments. We would prefer not to open the "Model Validation" section with that, but rather to keep it as its close, and recall it in the discussion, where we also suggest how to overcome this weakness.

We liked the idea to move the global MOC before the AMOC, but considering your comments #11, we replaced the global transport computed in depth space with the Southern Ocean MOC computed in density space. After this substitution, it makes sense to us to leave it after Atlantic and Indo-Pacific MOC results.

**11. In 3.2, the depth-space overturning means very little in the Southern Ocean. The global MOC should be calculated in density space. The Deacon cell (line 350) is not a physically relevant cell and it would be better to estimate the size of the lower overturning cell in density space.**

The use of potential density as the vertical coordinate, rather than depth, is more suitable for representing the MOC in the Southern Ocean, resulting in a better characterization of water mass transport. In particular, the wind-driven Deacon cell, which normally appears in depth-space MOC, is mostly an artefact of zonal and vertical integration at fixed depth level, and does not reflect any real-cross isopycnal flow. Following the Referee's comment, we replaced the global MOC in depth space with the computation in potential density space (where the potential density coordinate is referenced to 2000 db). The new plots for the global density-space MOC is shown below (Plot 1).

[Figure]

Plot 1. Meridional overturning circulation (in Sv) in potential density space (referenced to 2000 db, σ2) for the global domain. Solid lines represent positive (clockwise) contours. The contour interval is 2 Sv.

In the revisited manuscript, in order to better highlight the Southern Ocean circulation, we display a selected latitude range, from the southernmost boundary to 30ºS.

We added a paragraph to describe the new plot: "The MOC in depth-space is not the most suitable representation of the Southern Ocean overturning circulation. The Deacon cell, for example, is mostly due to a geometrical effect of the east-west slope of the isopycnals and no cross-isopycnal flow is associated with it (Döös and Webb 1994, Farneti et al. 2015). To account for a better characterization of water mass transports, the Southern Ocean MOC is presented in density space as a function of latitude and potential density σ2, referenced to the intermediate depth of 2000 m (Fig. 3d). Three primary cells are identified. The wind-driven subtropical cell is part of the horizontal subtropical gyres and is confined to the lightest density classes. This anticlockwise cell comprises a surface flow spreading poleward to 40S, compensated by an equatorward return flow. GLOB16 produces a subtropical cell of 18 Sv at 32ºS. Below, the upper cell is depicted by the large clockwise circulation, with a time-mean maximum value of 7 Sv. It mainly consists of upper circumpolar deep water that flows at depth southward to ~55ºS, upwells from 36.5 kg m$^{-3}$ to lighter density classes and returns northward as AAIW. The anticlockwise lower cell, in the densest layers, reaches 22 Sv and consists of the poleward lower circumpolar deep water and the deeper equatorward AABW. From 60ºS to the Antarctic continent, the transport represents the contribution of subpolar gyres in the Weddell and Ross Seas. Compared to GLOB16, the Southern Ocean MOC in the eddy-permitting configuration presents a stronger and more extended upper cell, but a slightly weaker transport in the subtropical cell, and an almost absent deep and dense flow in the lower cell (not shown)."

The new Fig 3 is reported below.

[Figure]

[Figure]

Plot 2. (Fig. 3 in the manuscript) Meridional overturning stream function (in Sv) averaged over the period 2009–2013, calculated in depth space for (a) the Atlantic and (c) the Indo-Pacific basins, in density space as function of $\sigma_2$ for (d) the Southern Ocean. The contour interval is 2 Sv in (a, d) and 3 Sv in (c). Thin solid lines represent positive (clockwise) contours; dashed lines represent negative (anticlockwise) contours. The stream functions were calculated with 0.5º latitudinal spacing to smooth out small-scale variations. (b) Time series of the AMOC at 26.5º N from RAPID observational estimates (blue), GLOB16 (red) and GLOB4 (black) numerical simulations.

**12. In Figures 2, 3, 5, 6, 8a,b, 9 and 10 there was no information on the GLOB4 results. However, occasionally, there were references in the text. As noted above, this manuscript will be much stronger if we know where and how improvements between GLOB4 and GLOB16 are manifested, so these results should be included wherever possible.**

We thank the referee for this comment. We agree that a more detailed comparison between GLOB16 and GLOB4 can definitely strengthen the manuscript. To present a more complete comparison with GLOB4, we modified subplots and/or add text in any relevant part of section 3, trying not to alter too much the manuscript structure or increase its length.

**13. In Fig. 5 I would also like to see a line indicating estimates and errors of each quantity from observations. (Some are listed, some are not). In particular, the Mozambique Channel transport is stated as being "within the range of observed estimates" without a reference!) Also, the ACC transport, listed as the average over all years, is steady and very low for the last 6 years XX it is this equilibrium value that should be listed, not the average over all years.**

Thanks for this comment, which helped to improve the plot. On Fig. 5, we added all the observed estimates with associated errors (when available) as reported in Table 2.

To better describe the variability of the ACC transport, we modified the related text in section 3.3 as follow "The zonal circumpolar transport drifts from a mean value of 131.2 Sv in 2004 to 117.3 Sv in 2013. The average volume transport is 122.6 (117.2) Sv over the 2004-2013 (2009-2013) period, lower but comparable to the recent observational estimate over the period 2007-2011 by Chidichimo et al. (2014)." Text describing the Mozambique Channel transport was probably not clear enough. We changed it in "The southward flux across the Mozambique Channel is 23.4 ± 5.4 (20.8 ± 5.8) in GLOB16 (GLOB4) and, for both models, follows within the broad compass of observed estimates, spanning from -29.1 Sv (DiMarco et al. 2002) to -16.7 ± 5.1 Sv (Ridderinkhof et al. 2010)."

**14. On line 444, it is ambiguous as to which "two transports are out of phase".**

We reworded this sentence as follows "Those transports vary out of phase with each other (Fig. 5d). When the flow is stronger through Fram Strait, it is weaker through Davis Strait and vice versa, indicating that the fluxes out of the Arctic Ocean across those straits partially balance each other."

**15. One open question which deserves more investigation is the lack of mesoscale variability in the Southern Ocean. There is a suggestion (in the Conclusion) that this is due to viscous parameterisations, but no quantitative information on what those parameterisations are. The Southern Ocean is one of the key locations where one might expect this resolution to make a**

**dynamical difference, but the very low variability and ACC transport indicate that something is missing here. I suggest a deeper quantitative comparison with other high resolution models is in order.**

Yes, the underestimation of Southern Ocean mesoscale variability reproduced in this GLOB16 simulation needed a deeper investigation. Unfortunately, re-running 10-year experiments is not faceable at the moment. To improve this aspect of GLOB16, we are currently performing short (2-year) sensitivity experiments, which includes a set of values of the coefficient for horizontal eddy viscosity. Preliminary results suggest improvements on both the ACC transport and mesoscale variability.

**We have made the following additional changes to the paper:**
In the list of References, we updated Stepanov et al. 2016, Storto et al. 2016.

We corrected the Mozambique Channel transport in Table 2 and accordingly added the following reference
Ridderinkhof, H., P. M. van der Werf, J. E. Ullgren, H. M. van Aken, P. J. van Leeuwen, and W. P. M. de Ruijter: Seasonal and interannual variability in the Mozambique Channel from moored current observations, J. Geophys. Res., 115, C06010, 2010.

**A 1/16° eddying simulation of the global NEMO sea ice-ocean system**

[revised manuscript text omitted]

**3.1 Mean temperature and salinity**

The mean fields of modelled potential temperature and salinity are here validated against  the EN3 (the UK Met Office Hadley Centre observational dataset, Ingleby and Huddleston 2007) climatology, both averaged over the same period 2009-2013. As expected, due to the temperature and salinity restoring applied at the ocean surface, the global mean SST and SSS biases are small (-0.06 for SST and -0.04 for SSS). There are weak cold biases in the tropics, extending over much of the subtropical band, with the  largest SST biases  (~1 °C warmer) collocated with positive SSS error (0.5–1.5 psu) over the western boundary currents in the Atlantic and North Pacific oceans (not shown). The overall pattern of surface biases is similar between the two models.

The surface biases of models forced by prescribed surface boundary conditions are, to a large degree, constrained by the forcing fields, but the analysis of subsurface fields allow for a stronger test of the model, revealing discrepancies in diapycnal mixing and advection pathways. The time- and zonal-average of modelled potential temperature and salinity are shown in Fig. 2 (a, b), along with their differences from EN3 (Fig. 2c, d). GLOB16 temperature field reproduces the expected large-scale features (Fig. 2a), with cold waters over all depths at high latitudes and warm water at shallow, low latitudes. GLOB16 salinity also follows expectation (Fig. 2b): the low salinity tongue (34.6 psu) of Antarctic Intermediate Water (AAIW), which sinks to ~1500 m depth between 60º-50ºS and propagates toward the equator; an high salinity (up to 35.2 psu) cell centred around 25ºS over the upper 300 m layer; a surface salinity minimum of 34.2 psu at 5º-10ºN connected to the strong precipitation in the inter-tropical convergence zone; high-salinity tongue associated with the Mediterranean Sea at about 35ºN; low-salinity water over the top 200 m north of 45ºN related to the Arctic melt water; and high-salinity (35.2 psu) water below 300 m depth north of 60ºN associated with the formation of cold, dense waters in the North Atlantic. All of these features are clearly present in the observation-based climatology (not shown).

The difference field for temperature (Fig. 2c) indicates that the modelled ocean is generally too warm at intermediate depth (100-300 m), with the exception of the AAIW, colder by 0.4 ºC. The largest differences, propagating down to 1000 m, are located in the northern hemisphere from ~40ºN (likely due to the Mediterranean Sea) poleward. The locations of the convective site set the positive and negative biases within the band 60-75ºN. Compared to EN3 temperature, the upper Arctic Ocean in GLOB16 is too warm (up to ~1.4 ºC at ~300 m), mainly due to a warmer Barents Sea inflow. The salinity field reproduced by GLOB16 differs from observations by ~0.15 psu at the most (Fig. 2d). Modelled and observed salinities agree well off Antarctica. The model is saltier by 0.1 psu at about 50ºS in the upper 400 m of the water column, and by 0.15 psu at the Equator at ~150 m. The model is too saline (up to 0.1 psu) between 200 and 600 m within the 45-55ºN latitude band, again likely related to the propagation of the Mediterranean overflow in the Atlantic Ocean. Conversely, it is 0.75 psu fresher in the top layer north of 60ºN. The differences between GLOB16 and climatologies for both fields are small below 1500 m depth. Although the overall biases are similar between the two model configurations in many latitude bands, there are some relevant differences (Fig. 2e, f). For instance, the Southern Ocean is generally warmer in GLOB4, with a larger positive salinity bias at ~400 m depth around 50°S. Both models are warm and saline in the above depth range in the northern mid and high latitudes, but the biases differ in magnitude and locations, highlighting the difference in path of the western boundary current. Both models are warmer than observations in the Arctic Ocean: the largest warming is confined in the upper 200 m depth in GLOB16, while the maximum, with a similar rate, is located between 300 and 500 m depth in GLOB4.

**3.2 Volume and heat transports**

Transports, in particular the meridional overturning circulation (MOC), are frequently used to evaluate the model performance. To provide an overview of the large-scale general circulation of the GLOB16 model, we present the time-mean meridional overturning stream function of the flow for a zonally averaged view. The MOC  is shown in Fig. 3, displayed in depth space for the Atlantic and the Indo-Pacific basins, and in density space for the Southern Ocean . In GLOB16, the Atlantic overturning (AMOC, Fig. 3a) reproduces the two overturning cells linked to the formation of North Atlantic Deep Water (NADW) and Antarctic Bottom Water (AABW).  The upper cell consists of northward surface flow in the top 1000 m, sinking north of 45° (with ~6 Sv sinking north of the Greenland Scotland Ridge), and a southward return flow mainly occurring between depths of ~1000 and ~3000 m. It reaches its maximum strength of ~20 Sv at a depth of 1000 m around 35ºN. An anticlockwise cell, associated with  AABW , fills the deep ocean below 3000 m, and reaches ~6 Sv. The cross-equatorial transport is ~16.5 Sv. At lower resolution, the overall transport in the Atlantic Ocean is reduced. The transport weakens in both the upper and lower cells, and the NADW flow extents much deeper as it flows southward, reaching ~3500 m at the equator (not shown).

Relevant measurements with respect to the mass transport in the Atlantic Ocean and the associated heat transport are provided by the RAPID/MOCHA program (e.g., Cunningham et al. 2007) that makes the net transport across 26.5°N available since spring 2004. Both models are in very good agreement with the RAPID observations at 26.5ºN. The GLOB16 overturning strength and variability, computed at that latitude for the simulated decade, is 20.1 ± 2.9 Sv, which is stronger than, but reasonably consistent with the RAPID estimates of 17.0 ± 3.6 Sv observed between April 2004 to December 2013 (McCarthy et al. 2015) (Table 2). The GLOB16 and RAPID mean values for the 2009-2013 period are 19.3 ± 3.1 and 15.6 ± 3.2, respectively (Table 1). In Fig. 3b, we compare the time series of the strength of the AMOC at 26.5ºN from the eddying model integration and the RAPID estimates. At that latitude, GLOB16 simulation realistically reproduces the AMOC temporal variability on seasonal and inter-annual time scales, although the simulated variability is lower than the observed. The high-resolution model misrepresents the two events of low AMOC observed in 2009 and 2010, when GLOB16 transport exhibits a clear, but much weaker than RAPID, decline. Time series
from the twin 1/4º simulation is also shown. The Atlantic overturning transport is generally
weaker in GLOB4, having a mean magnitude of 14.9 ± 2.6 Sv over the 10 simulated year,
~25% lower than the eddying model. GLOB4 underestimates RAPID values in the first
simulated years, closely follows RAPID from 2008, and does better capture the interannual
variability and the 2009-10 AMOC reductions. Stepanov et al. (under review) suggested that
the source of discrepancy between the two models in simulating the AMOC minima at 26.5ºN
might be related to the RAPID methodology used for the calculation, which does not fully take
into account the impact of the recirculation of the subtropical gyre on the mid-ocean transport.
Coarser resolution models, which cannot resolve processes near the western boundary, produce
weaker recirculation cell (e.g., Getzlaff et al. 2005, Roussenov et al. 2008, Zhang 2010).
Therefore, in GLOB4, a smaller impact of recirculation and eddies leads to a closer
correspondence between the model output and RAPID data. Table 1 shows that the good
agreement between GLOB16 and RAPID is true not only for the total AMOC transports, but
also for its components (the Florida Current, Ekman and the mid-ocean transports). Details on
the decomposition of the AMOC reproduced at 26.5ºN are given in Stepanov et al. (2016
).

The Indo-Pacific stream function with its intense equatorial upwelling is shown in Fig. 3c.
Apart from the uppermost layers, where Ekman transports dominate, the Indo-Pacific is filled
by the AABW cell that reaches its maximum values of ~18 Sv between 3000 and 4000 m
depth. As expected, the southward flow outcrops in the Northern Hemisphere consistently with
intermediate water formation and penetration of water from the circumpolar area near surface
and bottom, sandwiching a southward return flow at intermediate depths. Even though the
overall structure of the Indo-Pacific MOC does not differ much between the two models, the
different resolution corresponds to a ~30% decrease of the deep overturning (not shown).
~~directly wind-driven circulation is represented by a strong Deacon cell that peaks to ~27 Sv at~~
~ The MOC in depth-space is not the most suitable representation of
the Southern Ocean overturning circulation. The Deacon cell, for example, is mostly due to a
geometrical effect of the east-west slope of the isopycnals and no cross-isopycnal flow is
associated with it (Döös and Webb 1994, Farneti et al. 2015). To account for a better
characterization of water mass transports, the Southern Ocean MOC is presented in density
space as a function of latitude and potential density σ2, referenced to the intermediate depth of
2000 m (Fig. 3d). Three primary cells are identified. The wind-driven subtropical cell is part of
the horizontal subtropical gyres and is confined to the lightest density classes. This
anticlockwise cell comprises a surface flow spreading poleward to 40ºS, compensated by an
equatorward return flow. GLOB16 produces a subtropical cell of 18 Sv at 32ºS. Below, the
upper cell is depicted by the large clockwise circulation, with a time-mean maximum value of
7 Sv. It mainly consists of upper circumpolar deep water that flows at depth southward to

~55ºS, upwells from 36.5 kg m$^{-3}$ to lighter density classes and returns northward as AAIW. The anticlockwise lower cell, in the densest layers, reaches 22 Sv and consists of the poleward lower circumpolar deep water and the deeper equatorward AABW. From 60ºS to the Antarctic continent, the transport represents the contribution of subpolar gyres in the Weddell and Ross Seas. Compared to GLOB16, the Southern Ocean MOC in the eddy-permitting configuration presents a stronger and more extended upper cell, but a slightly weaker transport in the subtropical cell, and an almost absent deep and dense flow in the lower cell (not shown).

[revised manuscript text omitted]

**Code availability**

The NEMO model is freely available under the CeCILL public licence. After registration on the NEMO website (http://www.nemo-ocean.eu/), users can access the code (via Subversion, http://subversion.apache.org/) and run the model, following the procedure described in the "NEMO Quick Start Guide". The revision number of the code used for this study is 4510. The CMCC NEMOv3.4 code includes some additional modifications, applied to the base code. In particular, we modified the North Pole folding condition, introducing a more sophisticated optimization of the north fold algorithm (Epicoco et al. 2014), which leads to an extra increase in model performances (up to 20% time-reduction on the used architecture) without altering any physical process. The algorithm is now available in NEMO version 3.6. Interested readers can contact the authors for more information on the CMCC NEMOv3.4 code.

**Acknowledgements**

We thank the two anonymous reviewers for their thorough reading and constructive comments, which helped improving the manuscript. The financial support of the Italian Ministry of Education, University and Research, and Ministry for Environment, Land and Sea through the project GEMINA is gratefully acknowledged. We also acknowledge PRACE for awarding us the project ENSemble-based approach for global OCEAN forecasting (ENS4OCEAN) and providing access to resource on MareNostrum based at the Barcelona Supercomputing Center (Spain). The numerical results used here are available under request at CMCC. The RAPID data have been loaded from the following web pages: http://www.rapid.ac.uk/rapidmoc and http://www.rsmas.miami.edu/users/mocha. The EN3 subsurface ocean temperature and salinity data were collected, quality-controlled and distributed by the U.K. Met Office Hadley Centre. The Aviso altimeter products were produced by Ssalto/Duacs and distributed by Aviso, with support from CNES.

**References**

[revised manuscript text omitted]

Fig. 3. Meridional overturning stream function (in Sv) averaged over the period 2009–2013, calculated in depth space for (a) the Atlantic and (c) the Indo-Pacific basins, and in density space as function of $\sigma_2$ for (d) the Southern Ocean. The contour interval is 2 Sv in (a, d) and 3 Sv in (c). Thin solid lines represent positive (clockwise) contours; thick solid lines represent zero contours. The stream functions were calculated with 0.5º latitudinal spacing to smooth out small-scale variations. (b) Time series of the AMOC at 26.5º N from RAPID observational estimates (blue), GLOB16 (red) and GLOB4 (black) numerical simulations.

[Figure]

Fig. 4. (a) Time-mean Atlantic MHT (in PW) as a function of latitude. Red line is the total GLOB16 transport with its overturning (green) and gyre (dashed green) and components. Black line represents the total GLOB4 transport. Blue circles (squares) represent implied time-mean transport calculated by Large and Yeager 2009 (Trenberth and Fasullo 2008). Triangles indicate direct estimates with their uncertainty ranges from the 2009-2013 RAPID data (cyan), from Ganachaud and Wunsch 2003 (blue) and Lumpkin and Speer 2007 (magenta). (b) Times series of the total Atlantic MHT across 26.5° N as estimated by RAPID (blue), from GLOB16 (red) and GLOB4 (black).

[Figure]

Fig. 5. Time series of the monthly averaged volume transport (in Sv) of the (a) ACC, (b) ITF (decomposed in Timor passage (red), Ombai strait (blue) and Lombok strait (green)), through the (c) Mozambique Channel, (d) Bering Strait (black), Fram Strait (red) and Davis Strait (green), and (e) for dense overflow through Denmark Strait (black) and Faroe Bank Channel (red). Observed values with error bars (as reported in Table 2) are shown.

[Figure]

Fig. 6. (a) MLD (in m) averaged over March (in the Northern hemisphere) and September (in the Southern
hemisphere) 2009-2013 from (a) GLOB16, (b) GLOB4, and (c) the de Boyer Montégut et al. (2004) climatology,
based on a 0.03 threshold on density profiles. Model outputs are is shown on the native grid; observations are
interpolated on the eddy-permitting ORCA grid. Numbers of grid points are indicated on the axis, along with
indications of latitudes and longitudes.

[Figure]

[Figure]

Fig. 7. Time series of modelled MLD maxima (in km) in the North Atlantic Ocean (red), the Nordic Seas (blue) and the Southern Ocean (black) from GLOB16 (solid lines) and GLOB4 (dashed).

[Figure]

Fig. 8. (a) Mean GLOB16 seasonal cycles of sea ice extent ($10^6$ km$^2$) for the Arctic (black) and Antarctic (red) oceans compared to satellite observations (dashed line) provided by NSIDC. Sea ice extent is defined as the area enclosed in the 10% sea ice concentration contour. (b) Mean seasonal cycles of sea ice volume ($10^3$ km$^3$) for the Arctic Ocean (black) compared to PIOMAS reanalysis (dashed line), and for the Antarctica (red) compared to minimum and maximum values from ICESat. (c) Sea ice area export ($10^3$ km$^2$ month$^{-1}$) across Fram Strait for GLOB16 (red), GLOB4 (black) and observations (blue).

[Figure]

Fig. 9. Maximum (a, b) and minimum (c, d) Arctic sea ice concentration for the period 2009-2013 in GLOB16
(left) and observational data set (right).

[Figure]

Fig. 10. Maximum (a, b) and minimum (c, d) Antarctic sea ice concentration for the period 2009-2013 in
GLOB16 (left) and observational data set (right).

[Figure]

Fig. 11. Sea surface height variability (in m) from (a) the GLOB16 model, (b) the GLOB4 model and (c) AVISO. Modelled fields are shown on the own model grid; observations are interpolated on the eddy-permitting ORCA grid. Numbers of grid points are indicated on the axis, along with indications of latitudes and longitudes.

[Figure]

a)

b)

Fig. 12. (a) Latitudinal profiles of the global zonal-mean EKE (in cm² s⁻²) of the surface flow for 2013 from GLOB16 (red), GLOB4 (blue) and OSCAR (black). Scale is logarithmic. (b) As (a), but for the MKE of the surface flow (in cm² s⁻²).